# Provable Guarantees on Learning Hierarchical Generative Models with Deep CNNs

## Abstract

Learning deep networks is computationally hard in the general case. To show any positive theoretical results, one must make assumptions on the data distribution. Current theoretical works often make assumptions that are very far from describing real data, like sampling from Gaussian distribution or linear separability of the data. We describe an algorithm that learns convolutional neural network, assuming the data is sampled from a deep generative model that generates images level by level, where lower resolution images correspond to latent semantic classes. We analyze the convergence rate of our algorithm assuming the data is indeed generated according to this model (as well as additional assumptions). While we do not pretend to claim that the assumptions are realistic for natural images, we do believe that they capture some true properties of real data. Furthermore, we show that on CIFAR-10, the algorithm we analyze achieves results in the same ballpark with vanilla convolutional neural networks that are trained with SGD.

## 1 Introduction

The success of deep convolutional neural networks (CNN) has sparked many works trying to understand their behavior. As various theoretical studies have shown, learning deep networks is computationally hard in the general case, when no assumptions on the distribution of the data are taken (see for example Livni et al. (2014)). In practice, learning CNNs is done successfully using simple gradient-based optimization algorithms like SGD. Hence, to provide a theoretical analysis that will explain the practical success of deep learning, one must make assumptions on the distribution of the learned data. Currently, theoretical works in the literature of deep learning make rather strong assumptions, that clearly do not capture the properties of natural data. For example, many works assume that the examples are sampled from a Gaussian distribution, an assumption that is very far from describing distributions on natural images. Other works assume linear separability of the data, which clearly does not hold for any rich enough dataset.

In this work, we assume the data is generated from a deep generative model. According to this model, the examples are generated in a hierarchical manner: each example (image) is generated by first drawing a high-level semantic image, and iteratively refining the image, each time generating a lower-level image based on the higher-level semantics from the previous step. Similar models were suggested in other works as good descriptions of natural images encountered in real world data. While we do not claim that natural images actually come from such distribution, we believe that it captures some key properties of real world data. Importantly, the problem we study is not trivially learned by simple "shallow" learning algorithms.

Our work analyzes the training of a CNN on data from this generative distribution. In a shallow case, where the generative model has only two levels of hierarchy, we analyze the behavior of standard gradient-descent. In the deep case, where our model can have many levels of hierarchy, we analyze a layerwise optimization algorithm, proving its convergence under the assumed generative model (as well as additional, admittedly strong, assumptions). The algorithm we analyze is somewhat different than optimization algorithms that are commonly used in practice, and may seem "tailored" to solve the problem of learning our generative heirarchical model. We show that implementing this algorithm to learn real-world data (CIFAR-10 dataset) achieves performance that are in the same ballpark as a vanilla CNN trained with SGD-based optimizer. This result hints that our model and algorithm indeed capture properties of distributions and algorithms that are common in practice.

## 2 RELATED WORK

Any theoretical work that aims to give positive results on learning deep networks must make assumptions on the data distribution and on the learning algorithm. We can roughly divide such works into three categories: (1) works that study practical algorithms (SGD) solving "simple" problems that can be otherwise learned with "shallow" algorithms. (2) works that study problems with less restrictive assumptions, but using algorithms that are not applicable in practice. (3) works that study a generative model similar to ours, but either give no theoretical guarantees, or otherwise analyze an algorithms that are not applicable for learning CNNs on image data.

Trying to study a practically useful algorithm, Daniely (2017) proves that SGD learns a function that approximates the best function in the conjugate kernel space derived from the network architecture. Although this work provides guarantees for a wide range of deep architectures, there is no empirical evidence that the best function in the conjugate kernel space performs at the same ballpark as CNNs. The work of Andoni et al. (2014) shows guarantees on learning low-degree polynomials, which is learnable via SVM or direct feature mapping. Other works study shallow (one-hidden-layer) networks under some significant assumptions. The works of Gori & Tesi (1992); Brutzkus et al. (2017) study the convergence of SGD trained on linearly separable data, which could be learned with the Perceptron algorithm, and the works of Brutzkus & Globerson (2017); Tian (2017); Li & Yuan (2017); Zhong et al. (2017) assume that the data is generated from Gaussian distribution, an assumption that clearly does not hold in real-world data. The work of Du et al. (2017) extends the results in Brutzkus & Globerson (2017), showing recovery of convolutional kernels without assuming Gaussian distribution, but is still limited to the regime of shallow two-layer network.

Another line of work aims to analyze the learning of deep architectures, in cases that exceed the capacity of shallow learning. The works of Livni et al. (2014); Zhang et al. (2015; 2016a) show polynomial-time algorithms aimed at learning deep models, but that seem far from performing well in practice. The work of Zhang et al. (2016b) analyses a method of learning a model similar to CNN which can be applied to learn multi-layer networks, but the analysis is limited to shallow two-layer settings, when the formulated problem is convex.

Finally, there have been a few works suggesting distributional assumptions on the data that are similar in spirit to the generative model that we analyze in this paper. Again, these works can be largely categorized into two classes: works that provide algorithms that have theoretical guarantees but are not applicable for learning CNNs, and works that show practical results without theoretical guarantees. The work of Arora et al. (2014) shows a provably efficient algorithm for learning a deep representation, but this algorithm seems far from capturing the behavior of algorithms used in practice. Our approach can be seen as an extension of the work of Mossel (2016), who studies Hierarchal Generative Models, focusing on algorithms and models that are applicable to biological data. Mossel (2016) suggests that similar models may be used to define image refinement processes, and our work shows that this is indeed the case, while providing both theoretical proofs and empirical evidence to this claim. Finally, the works of Tang et al. (2012); Patel et al. (2016); Van den Oord & Schrauwen (2014) study generative models similar to ours, with promising empirical results when implementing EM inspired algorithms, but giving no theoretical foundations whatsoever.

## 3 DISTRIBUTIONAL ASSUMPTIONS

In this paper, we are concerned with the problem of learning binary classification of images. Assume we are given a sample $S = \{(\boldsymbol{X}_1, y_1), \ldots, (\boldsymbol{X}_n, y_n)\}$ from some distribution $\mathcal{D}$ on $\mathbb{R}^{m \times m} \times \mathcal{Y}$, where $\mathcal{Y} = \{\pm 1\}$ is our labels. We wish to learn a CNN model that will classify the images correctly. As noted, if we hope to give any guarantees on the success of learning such model, we must make strong assumptions on the distribution $\mathcal{D}$. One naive assumption on $\mathcal{D}$ is that it is linearly separable: there exists $\boldsymbol{W}^* \in \mathbb{R}^{m \times m}$ such that $y \langle \boldsymbol{W}^*, \boldsymbol{X} \rangle \geq 1$ for $(\boldsymbol{X}, y) \sim \mathcal{D}$, where we denote $\langle \boldsymbol{A}, \boldsymbol{B} \rangle = \mathrm{tr}(\boldsymbol{A}^\top \boldsymbol{B})$. Clearly, such assumption does not hold for any rich enough distribution of natural images (as linear classifiers give poor results on natural image datasets).

To provide a more realistic assumption, we will instead assume the following:

**Assumption 1** *The images in $\mathcal{D}$ are generated from a latent distribution $\mathcal{G}_0$, such that $\mathcal{G}_0$ is linearly separable.*

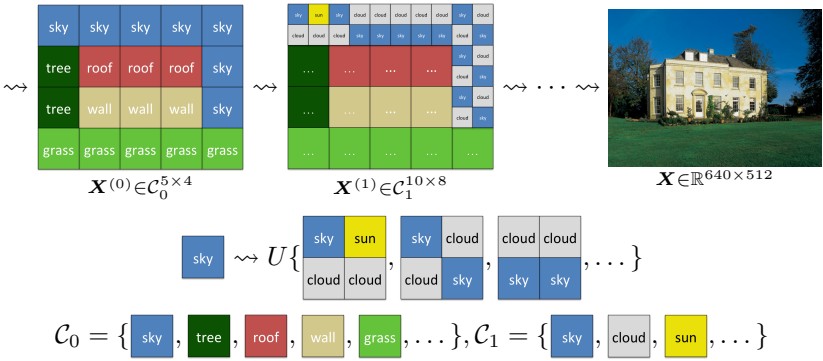

Figure 1: Generative model schematic description

Thus, sampling an image $(\boldsymbol{X}, y) \sim \mathcal{D}$ is equivalent to sampling a latent representation $(\boldsymbol{X}^{(0)}, y) \sim \mathcal{G}_0$, and then sampling $(\boldsymbol{X}, y) \sim \mathcal{G}_{\boldsymbol{X}^{(0)}}$, where $\mathcal{G}_{\boldsymbol{X}^{(0)}}$ is a distribution dependent on $\boldsymbol{X}^{(0)}$. Note that in this case, the distribution $\mathcal{D}$ can be very complex, although the images sampled from it have a latent representation that is relatively simple. To give a concrete description of the latent representation, we will assume that $\mathcal{G}_0$ is a distribution over "semantic images": images where each "pixel" represents a semantic class. Formally, we will assume that $\boldsymbol{X}^{(0)} \in \mathcal{C}_0^{m_0 \times m_0}$, where $\mathcal{C}_0$ is some finite set of "semantic classes".

Now, we want to describe the generative process of generating the image $\boldsymbol{X}$ from the semantic representation $\boldsymbol{X}^{(0)}$. This process is done in a hierarchical manner: Given the high-level semantic representation $\boldsymbol{X}^{(0)}$, where each "pixel" represents a semantic class (for example, background, sky, grass etc.), we generate a lower level image $\boldsymbol{X}^{(1)}$, where each patch comes from a distribution depending on each "pixel" of the high-level representation, generating a larger semantic image (lower level semantic classes for natural images could be: edges, corners, texture etc.). We can repeat this process iteratively any number of times, each time creating a larger image of lower level semantic classes, thus generating a sequence of increasingly large semantic images $\boldsymbol{X}^{(0)}, \boldsymbol{X}^{(1)}, \ldots, \boldsymbol{X}^{(d)}$. Since $\mathcal{D}$ is a distribution over greyscale images, we will assume that the last iteration of this process samples patches over $\mathbb{R}$, i.e $\boldsymbol{X} = \boldsymbol{X}^{(d)} \in \mathbb{R}^{m \times m}$. This model is described schematically in figure 1, with a formal description given in section 3.1. In section 3.2 we describe a synthetic example of digit images generated according to this model.

## 3.1 Formal Description

Here, we will give a detailed description of the generative process described above. To generate an example, we start by sampling the label $y \sim U(\mathcal{Y})$, where $U(\mathcal{Y})$ is the uniform distribution over the set of labels. Given $y$, we generate a small image with $m_0 \times m_0$ pixels, where each pixel belongs to a set $\mathcal{C}_0$. Elements of $\mathcal{C}_0$ corresponds to semantic entities (e.g. "sky", "grass", etc.). The generated image, denoted $\boldsymbol{X}^{(0)} \in \mathcal{C}_0^{m_0 \times m_0}$, is sampled according to some simple distribution $\mathcal{D}_y$. Next, we generate a new image $\boldsymbol{X}^{(1)} \in \mathcal{C}_1^{m_0 s \times m_0 s}$ as follows. Pixel $i$ in $\boldsymbol{X}^{(0)}$ corresponds to some $c \in \mathcal{C}_0$. For every such $c$, let $\mathcal{D}_c$ be the uniform distribution over a finite set of patches $S_c^{(0)} \subset \mathcal{C}_1^{s \times s}$, where we refer to $s$ as a "patch size". We assume the following:

**Assumption 2** *The sets $\{S_c^{(0)}\}_{c \in \mathcal{C}_0}$ are disjoint, and each one is of size $k$.*

So, pixel $i$ in $\boldsymbol{X}^{(0)}$ whose value is $c \in \mathcal{C}_0$ generates a patch of size $s$ in $\boldsymbol{X}^{(1)}$ by sampling the patch according to $\mathcal{D}_c$. This process continues, yielding images $\boldsymbol{X}^{(2)}, \ldots, \boldsymbol{X}^{(d)}$ whose sizes are $m_0 s^2 \times m_0 s^2, \ldots, m_0 s^d \times m_0 s^d$. Each pixel in level $i$ comes from $\mathcal{C}_i$, and each patch comes from $S_c^{(i-1)}$ for some $c \in \mathcal{C}_{i-1}$. We assume that $\mathcal{C}_d = \mathbb{R}$, so the final image is over the reals. The observed example is the pair $(\boldsymbol{X}^{(d)}, y)$. We denote the distribution generating the image of level $i$ by $\mathcal{G}_i$.

As noted, we assume that $\mathcal{G}_0$, the distribution that generates the high-level semantic images, is linearly separable. As $\mathcal{G}_0$ is a distribution over semantic images, i.e over $\mathcal{C}_0^{m \times m} \times \mathcal{Y}$, we need to define what "linear separability" means in this case. To do this, we assign to the $i$-th class in $\mathcal{C}_0$ the unit vector $e_i \in \mathbb{R}^{|\mathcal{C}_0|}$. Then we represent $\boldsymbol{X} \in \mathcal{C}_0^{m \times m}$ with an equivalent tensor $\mathbf{X} \in \mathbb{R}^{|\mathcal{C}_0| \times m \times m}$, and define linear separability with respect to the tensor representation. Namely, there exists $\mathbf{W}^*$ such that for $(\boldsymbol{X}, y) \sim \mathcal{G}_0$ it holds that $y \langle \mathbf{X}, \mathbf{W}^* \rangle \geq 1$, where $\langle \mathbf{X}, \mathbf{W}^* \rangle := \sum_{i,j,k} X_{i,j,k} \cdot W_{i,j,k}^*$.

Notice that our semantic classes partition the patches in the image into disjoint sets. There are various ways to partition the patches, some of which might in fact generate the same distribution. For the analysis, we add an assumption that ensures that the semantic classes defined in the model are different enough from each other. We identify each class with a matrix that captures the frequency of its appearance in the image, with respect to the label of the image. For this, we define the "labeled frequency matrix" of class $c \in \mathcal{C}_i$ to be the matrix $\boldsymbol{F}_c \in \mathbb{R}^{m_0 s^i \times m_0 s^i}$ defined as follows: for every $c$ we denote by $\mathcal{I}_c$ the operator that takes a matrix as its input and replaces every element of the input by the boolean that indicates whether it equals to $c$. Then, we define: $\boldsymbol{F}_c := \mathbb{E}_{(\boldsymbol{Z},y) \sim \mathcal{G}_i} [y \mathcal{I}_c(\boldsymbol{Z})]$. Notice that $\boldsymbol{F}_c$ is the "mean" image over the distribution for every given semantic class $c$. For example, semantic classes that tend to appear in the upper-left corner of the image for positive images will have positive values in the upper-left entries of $\boldsymbol{F}_c$. We now assume:

**Assumption 3** *The vectors $\{\boldsymbol{F}_c\}_{c \in \mathcal{C}_i}$ are linearly independent in pairs.*

For each $c_1, c_2 \in \mathcal{C}_i$ we denote the angle between $\boldsymbol{F}_{c_1}$ and $\boldsymbol{F}_{c_2}$ by: $\angle(\boldsymbol{F}_{c_1}, \boldsymbol{F}_{c_2}) := \arccos\left(\frac{\langle \boldsymbol{F}_{c_1} \boldsymbol{F}_{c_2} \rangle}{\|\boldsymbol{F}_{c_1}\|_F \|\boldsymbol{F}_{c_2}\|_F}\right)$. Denote $\theta := \min_{i,c_1,c_2 \in \mathcal{C}_i} \angle(\boldsymbol{F}_{c_1}, \boldsymbol{F}_{c_2})$ and $\lambda := \min_{i,c \in \mathcal{C}_i} \|\boldsymbol{F}_c\|_F$. From the linear independence assumption it follows that both $\theta$ and $\lambda$ are strictly positive. The runtime of the algorithms described in the next sections depend on $1/\theta$ and $1/\lambda$.

## 3.2 SYNTHETIC DIGITS EXAMPLE

To demonstrate our generative model, we use a small synthetic example to generate images of digits. In this case, we use a three levels model, where semantic classes represent lines, corners etc. In the notations above, we use:

$$\mathcal{C}_0 = \{\ \square, \boxplus, \boxminus, \boxminus, \boxminus, \boxminus, \boxminus, \boxplus, \boxdot\ \}\ ,\ \mathcal{C}_1 = \{\ \square, \boxslash, \boxminus, \odot, \boxminus, \boxdot\ \}\ ,\quad \mathcal{C}_2 = \mathbb{R}$$

We define $\mathcal{D}_{even}, \mathcal{D}_{odd}$ to be the uniform distributions over the even/odd digital representations:

$$\mathcal{D}_{even} = U\{\ \boxed{0}\ ,\ \boxed{2}\ ,\ \boxed{4}\ ,\ \boxed{6}\ ,\ \boxed{8}\ \}, \mathcal{D}_{odd} = U\{\ \boxed{1}\ ,\ \boxed{3}\ ,\ \boxed{5}\ ,\ \boxed{7}\ ,\ \boxed{9}\ \}$$

Now, in the second level of the generative model, each pixel in $\mathcal{C}_0$ can generate one of four possible manifestations. For example, for the pixel $\boxed{\urcorner}$, we sample over: $\boxed{\ }\ ,\ \boxed{\ }\ ,\ \boxed{\ }\ ,\ \boxed{\ }$ . Similarly, in the final level we sample for each $c \in \mathcal{C}_1$ from a distribution $\mathcal{D}_c$ supported over 4 elements. For example, for the pixel $\boxed{\mid}$, we sample over: $\blacksquare\ ,\ \blacksquare\ ,\ \blacksquare\ ,\ \blacksquare$ .

Notice that though this example is extremely simplistic, it can generate $4^9$ examples per digit in the first level, and $4^{90}$ examples for each digit in the final layer, amounting to $9 \cdot 4^{90} \approx 1.38 \cdot 10^{55}$ different examples. Figure 2 shows the process output.

## 3.3 LINEAR SEPARABILITY

While the generative model described above seems complex, one might wonder whether this model, along with all the distributional assumptions given so far, is in fact an overly complicated fashion to describe a rather simple problem. While generally speaking, it is not clear how to measure the "complexity" of the suggested distribution, we can at least address the question of whether this distribution is linearly separable or not.

It is immediate to show that when all the generated patches are orthogonal to each other, this distribution becomes linearly separable. We analyze this case in the next section, and prove some results in this simplistic case. On the other hand, when such assumption is not taken, we argue that the model is typically far from being linearly separable. To show this, we generate examples using this

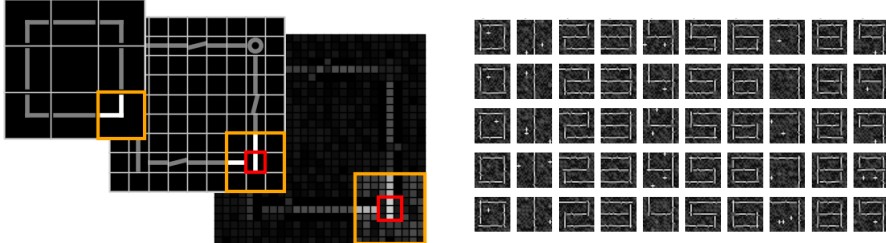

Figure 2: Left: Image generation process example. Right: Synthetic examples generated.

| Experiment | $\theta$ | $\lambda$ | Linear | CNN | Ours |
|---|---|---|---|---|---|
| $k = 30$ | 0.96 | 0.17 | 0.65 | 1.0 | 1.0 |
| $k = 40$ | 1.64 | 0.16 | 0.58 | 1.0 | 1.0 |
| $k = 50$ | 1.65 | 0.18 | 0.58 | 0.99 | 1.0 |

Figure 3: Linear and CNN classifiers on generated data

generative distribution, where the generated patches are chosen randomly. Recall that learning a linear separator is a convex problem. Therefore, if the generated data was linearly separable, finding such a separator would be trivial, using the SGD algorithm. Figure 3 shows the accuracy of the linear classifier, compared to the performance of a simple CNN and our algorithm. As is clear from these results, a linear separator achieves very poor results on this data. Furthermore, to show that the generated distribution does not break Assumption 3, we show the values of $\theta$ and $\lambda$, as measured on the generated data.

The exact details of the experiment are given in appendix D.

## 4 TWO-LEVEL MODEL

As a warm-up, we first limit ourselves to observing distributions with only two levels in the hierarchy. Hence, the process of generating examples is simply: $\rightsquigarrow^{\mathcal{G}_0} (\boldsymbol{X}^{(0)}, y) \rightsquigarrow^{\mathcal{G}_1} (\boldsymbol{X}^{(1)}, y)$. Furthermore, in this section we add an important assumption on the data distribution:

**Assumption 4** *The set of patches that compose the images generated by $\mathcal{G}_1$ is orthonormal. Recall that we denoted $S_c^{(0)} \subset \mathcal{C}_0^{s \times s}$ the set of patches that are sampled for each pixel of class $c$. Our assumption means that $\forall \boldsymbol{P}_1, \boldsymbol{P}_2 \in \{S_c^{(0)}\}_{c \in \mathcal{C}_0}$ we have $\langle \boldsymbol{P}_1, \boldsymbol{P}_2 \rangle = \mathbf{1}_{\boldsymbol{P}_1 = \boldsymbol{P}_2}$.*

Notice that this assumption implies the following lemma:

**Lemma 1** *Given assumptions 1, 2, 4, $\mathcal{G}_1$ is linearly separable with margin $\sqrt{k}\|\mathbf{W}^*\|$, where $\mathbf{W}^*$ is the linear separator of $\mathcal{G}_0$ (see section 3.1). Consequently, running linear SVM finds a classifier with zero classification loss on this distribution in $O(k\|\mathbf{W}^*\|^2)$ iterations.*

While usually in classification problems our goal is to minimize the training loss, in this case we would like to find an algorithm that instead recovers the latent representation $\boldsymbol{X}^{(0)}$ given the raw image $\boldsymbol{X}^{(1)}$. The reasons for this requirement will become clear in the next section. We will prove that training a two layer linear CNN with gradient-descent (GD) on distribution $\mathcal{G}_1$ implicitly recovers the latent representation, even if $\mathcal{G}_0$ is not linearly separable. Note that this is a rather surprising result, as we are only observing $(\boldsymbol{X}^{(1)}, y)$, and have no access to the latent representation $\boldsymbol{X}^{(0)}$. The following section will give the details of this result: in section 4.1 we describe the details of the GD algorithm, and in section 4.2 we describe the theoretical analysis of this algorithm.

## 4.1 Algorithm: GD on Two-Layer CNN

Recall that for $(\boldsymbol{X}, y) \sim \mathcal{G}_1$, $\boldsymbol{X}$ is an image of size $sm_0 \times sm_0$, that is composed of $s \times s$ patches generated from a high-level semantic image of size $m_0 \times m_0$. To simplify notation, we consider $\boldsymbol{X}$ to be "reshaped" such that each column is a patch in the original image (the so called "im2col" operation), so $\boldsymbol{X} \in \mathbb{R}^{s^2 \times m_0^2}$. Given $\boldsymbol{K} \in \mathbb{R}^{s^2 \times n}, \boldsymbol{W} \in \mathbb{R}^{m_0^2 \times n}$, we define a convolutional subnet to be a function $\mathcal{N}_{\boldsymbol{K}, \boldsymbol{W}} : \mathbb{R}^{s^2 \times m_0^2} \to \mathbb{R}$ such that: $\mathcal{N}_{\boldsymbol{K}, \boldsymbol{W}}(\boldsymbol{X}) = \langle \boldsymbol{W}^\top, \boldsymbol{K}^\top \boldsymbol{X} \rangle$.

This is equivalent to a convolution operation on an image, followed by a linear weighted sum: multiplying $\boldsymbol{X}$ by $\boldsymbol{K}^\top$ is equivalent to performing a non-overlapping convolution operation with kernels $\boldsymbol{k}_1, \ldots, \boldsymbol{k}_n$ on the original image (where $\boldsymbol{k}_i$ is the $i$-th vector of matrix $\boldsymbol{K}$). Flattening the resulting matrix and multiplying by the weights in $\boldsymbol{W}$ yields the second linear layer.

The top linear layer of the network outputs a prediction for the label $y \in \mathcal{Y}$. The network is trained with respect to the loss $\mathcal{L}_{\boldsymbol{K}, \boldsymbol{W}}^S$ on a given set of examples $S$, defined as the expected value of some label dependent loss function $\ell_y : \mathbb{R} \to \mathbb{R}$

$$\mathcal{L}_{\boldsymbol{K}, \boldsymbol{W}}^S = \mathbb{E}_{(\boldsymbol{X}, y) \sim S} \left[ \ell_y(\mathcal{N}_{\boldsymbol{K}, \boldsymbol{W}}(\boldsymbol{X})) \right]$$

For the analysis, we use the loss $\ell_y(\hat{y}) = -y\hat{y}$. This loss simplifies the analysis, and seems to capture a similar behavior to other loss types used in practice.

Although in practice we perform a variant of SGD on a sample of the data to train the network, we perform the analysis with respect to the population loss: $\mathcal{L}_{\boldsymbol{K}, \boldsymbol{W}} = \mathbb{E}_{(\boldsymbol{X}, y) \sim \mathcal{G}} \left[ \ell_y(\mathcal{N}_{\boldsymbol{K}, \boldsymbol{W}}(X)) \right]$. We denote $\boldsymbol{K}_t$ the weights of the first layer of the network in iteration $t$, and denote $\boldsymbol{W}_0$ the initial weights of the second layer. For simplicity of the analysis, we assume that only the first layer of the network is trained, while the weights of the second layer are fixed. Thus, we perform the following update step at each iteration of the gradient descent: $\boldsymbol{K}_t = \boldsymbol{K}_{t-1} - \eta \frac{\partial}{\partial \boldsymbol{K}} \mathcal{L}_{\boldsymbol{K}_{t-1}, \boldsymbol{W}_0}$.

We initialize $\boldsymbol{W}_0 \sim \mathcal{N}(0, 1)$ and each $n$-dimensional column of $\boldsymbol{K}_0$ is sampled from the uniform distribution on the sphere of radius $\frac{\sigma}{2\sqrt{n}}$, where $\sigma$ is a parameter of the algorithm.

## 4.2 Theoretical Analysis

In this section, we will consider a more general case, where we do not assume that $\mathcal{G}_0$ is linearly separable (Assumption 1). As noted, our main claim in this section is that training the two-layer Conv net implicitly recovers the latent semantic representation of the image. Specifically, we show that gradient-descent finds an embedding of the observed patches into a space such that patches from the same semantic class are close to each other, while patches from different classes are far. Note that this property is indeed enough to recover the latent representation, as running a trivial clustering algorithm on the embedded patches would reconstruct the latent image. Recall that we do not have access to the latent distribution, and thus cannot possibly learn such embedding directly. This surprising property of gradient descent is the key feature that allows our main algorithm (described in the next section) to learn the high-level semantics of the images. Our claim is given in the following theorem:

**Theorem 1** *Suppose that assumptions 2, 3, 4 hold. Let $\theta, \lambda$ as described in section 3.1. Assume we train a two-layer network of size $n > \frac{2\pi}{0.23\theta} \log(\frac{|\mathcal{C}|}{\delta})$ with respect to the population loss on distribution $\mathcal{G}_1$, with learning rate $\eta$, for $T > \frac{(\gamma + 2\sigma)k}{\eta \lambda}$ iterations, for some $\gamma > 0$. Assume that the training is as described in section 4.1, where the parameter $\sigma$ of the initialization is also described there. Then with probability of at least $1 - \delta$:*

1. *for each $c \in \mathcal{C}_0$, for every $x_1, x_2 \in S_c^{(0)}$ we get $\|\boldsymbol{K}_T^\top \cdot x_1 - \boldsymbol{K}_T^\top \cdot x_2\| < \sigma$*

2. *for $c_1, c_2 \in \mathcal{C}_0$, if $c_1 \neq c_2$, for every $x_1 \in S_{c_1}^{(0)}, x_2 \in S_{c_2}^{(0)}$, we get $\|\boldsymbol{K}_T^\top \cdot x_1 - \boldsymbol{K}_T^\top \cdot x_2\| > \gamma$*

We give a similar analysis for the SGD algorithm in appendix B.

## 5    DEEP MODEL

Now, assume we are given data from a deep generative distribution as described in section 3 (with $d \geq 2$), and our goal is to learn a classifier that predicts the label for each image. A reasonable approach, given the properties of the above distribution, would be to try to infer from the raw image the higher-level semantic representations. If given an example $(\boldsymbol{X}, y)$ we succeed to infer the sequence that generated it, $\boldsymbol{X}^{(0)}, \boldsymbol{X}^{(1)}, \ldots, \boldsymbol{X}^{(d)}$, we could then use SVM on the high-level representation, and learn to infer its label.

Given the results of the previous section, we can recover these semantic representations by training a two-layer Conv net and applying a trivial clustering on the resulting patches. Note that in general we do not assume that all the patches are orthogonal (Assumption 4), as such a property will make the whole model linearly separable. Thus, to apply the results of section 4.2, we would use the clustering algorithm both for standard clustering and also as an "orthogonalization" step, that will map different clusters to orthonormal vectors.

We next describe in details the full algorithm (section 5.1), and show that it finds a network that achieves zero loss in polynomial time (section 5.2).

### 5.1    ALGORITHM DESCRIPTION

The algorithm we suggest is built from three building-blocks composed together to construct the full algorithm: (1) clustering algorithm, (2) gradient-based optimization of two-layer Conv net and (3) a simple classification algorithm. In order to expose the latent representation of each layer in the generative model, we perform the following iteratively:

**(1)** Run a k-means algorithm on patches of size $s \times s$ from the input images defined by the previous step (or the original images in the first step), w.r.t. the cosine distance, to get $k_i$ cluster centers.
**(2)** Run a convolution operation with the cluster centroids as kernels, followed by ReLU with a fixed bias and a pooling operation. This will result in mapping the patches in the input images to (approximately) orthogonal vectors in an intermediate space $\mathbb{R}^{k_i}$.
**(3)** Initialize a 1x1 convolution operation, that maps from $k_i$ channels into $n$ channels, followed by a linear layer that predicts $\mathcal{Y}$. We train this two-layer subnet using gradient-descent. As an immediate corollary of the analysis in the previous section, this step implicitly learns an embedding of the patches into a space where patches from the same semantic class are close to each other, while patches from different classes are far away. This lays the groundwork for the clustering step of the next iteration.
**(4)** "Throw" the last linear layer, thus leaving a trained block of $\text{Conv}(s \times s)\text{-ReLU-Pool-Conv}(1 \times 1)$ which finds a "good" embedding of the patches of the input image, and repeat the process again, where the output of this block is the input to step 1.

Finally, after we perform this process for $d$ times, we get a network of depth $d$ composed from $\text{Conv}(s \times s)\text{-ReLU-Pool-Conv}(1 \times 1)$ blocks. Then, we feed the output of this (already trained) network to an SVM algorithm that learns a linear classifier, training it to infer the label $y$ from the semantic representation that the convolution network outputs. We now describe the building blocks for the algorithm, followed by the definition of the complete algorithm.

**Clustering**. The first block of the algorithm is the clustering step. We denote $\text{KMEANS}(S, k)$ to be the output of the k-means++ algorithm (Arthur & Vassilvitskii (2007)) running on sample $S$ to find $k$ clusters, w.r.t the cosine similarity. We assume the algorithm returns a mapping $\phi_S : \mathbb{R}^s \to \mathbb{R}^k$, such that if $\boldsymbol{x}_i, \boldsymbol{x}_j \in S$ are in the same cluster then $\phi_S(\boldsymbol{x}_i) = \phi_S(\boldsymbol{x}_j)$, and otherwise $\phi_S(\boldsymbol{x}_i) \perp \phi_S(\boldsymbol{x}_j)$.

For the consistency with common CNN architecture, we can use the centroids of each cluster as kernels for a convolution operation. Combining this with ReLU with a fixed bias and a pooling operation gives an operation that maps each patch to a single vector, where vectors of different patches are approximately orthogonal.

**Two-Layer Network Algorithm** We denote $\text{TLGD}(S, T, \eta, n, \sigma)$ the GD algorithm, training a two-layer Conv net of width $n$, on sample $S$, with learning rate $\eta$ for $T$ interations. This algorithm returns the weights of the first layer $\boldsymbol{K}_T$. The details of this algorithm are described in section 4.1.

**Classification Algorithm** Finally, the last building block of the algorithm is a classification stage, that is used on top of the deep convolution architecture learned in the previous steps. As we show, at this stage the examples are linearly separated, so we can use the SVM algorithm to find large margin linear separator. We denote $\text{SVM}(S)$ the output of running SVM on sample $S$.

**Complete Algorithm** Utilizing the building blocks described previously, our algorithm learns a deep CNN layer after layer. This network is used to infer the label for each image. The algorithm is described formally in algorithm 1. In the description, we use the notation $\phi * \mathbf{A}$ to denote the operation of applying a map $\phi : \mathcal{K}_0^{m_0} \to \mathcal{K}_1^{m_1}$ on a tensor $\mathbf{A}$, replacing patches of size $m_0$ by vectors in $\mathcal{K}_1^{m_1}$. Formally: $\phi * \mathbf{A} := [\phi(\mathbf{A}_{:,i \cdot m_0 \ldots (i+1) \cdot m_0})]_i$

---

**Algorithm 1** Deep Layerwise Clustering

**input**:
  numbers $\eta, T, n, \sigma, k_1, \ldots, k_d$
  sample $S = \{(\boldsymbol{X}_1, y_1), \ldots (\boldsymbol{X}_N, y_N)\} \subseteq \mathbb{R}^{m_0 s^d \times m_0 s^d} \times \mathcal{Y}$
$h_d \leftarrow id$
**for** $i = d \ldots 1$ **do**
  set $S_i \leftarrow \{(h_i(\boldsymbol{X}_1), y_1), \ldots, (h_i(\boldsymbol{X}_N), y_N)\}$ // construct sample using the current network
  set $P_i \leftarrow \{\text{patches of size s} \times \text{s from } S_i\}$ // generate patches from the current sample
  set $\phi_i \leftarrow \text{KMEANS}(P_i, k_i)$ // cluster the sampled patches
  set $\hat{S}_i \leftarrow \{(\phi_i * h_i(\boldsymbol{X}_1), y_1), \ldots, (\phi_i * h_i(\boldsymbol{X}_N), y_N)\}$ // map patches to orthogonal vectors using $\phi_i$
  **if** $i > 1$ **then**
    set $K_{i-1} \leftarrow \text{TLGD}(\hat{S}_i, T, \eta, n, \sigma)$ // train a two-layer net to find a "good" embedding for the patches
    set $h_{i-1} \leftarrow K_{i-1}^{\top}(\phi_i * h_i)$ // add the current block to the network
  **end if**
**end for**
set $h \leftarrow \text{SVM}(\hat{S}_0)$
return $h \circ h_1$

---

## 5.2 THEORETICAL ANALYSIS

In this section we show that, algorithm 1 learns in polynomial time (with high probability) a network model that correctly classifies the examples according to their labels.

Recall that we have shown that training a two-layer network implicitly learns an embedding of the patches into a space where patches from the same semantic class are close to each other, while patches from different classes are far. Using this, we show that performing clustering + two-layer network training iteratively, layer by layer, leads to revealing the underlying model, and hence the algorithm's convergence. We claim that algorithm 1 successfully learns a model that correctly classifies the examples sampled from the observed distribution $\mathcal{G}_d$. This is stated in the following theorem:

**Theorem 2** *Suppose that assumptions 1, 2, 3 hold. Fix $\delta > 0$, and let $\delta' = \frac{\delta}{2d}$. Denote $C := \max_{i<d} |\mathcal{C}_i|$ the maximal number of semantic classes in any level $i$. Choose $\sigma = \frac{1}{s}$, $n > \frac{2\pi}{0.23\theta} \log(\frac{C}{\delta'})$, $T > \frac{(C/\sqrt{\delta'} + 2\sigma)k}{\eta\lambda}$, $k_1 = |\mathcal{C}_0|, \ldots, k_{d-1} = |\mathcal{C}_{d-2}|, k_d = k|\mathcal{C}_{d-1}|$. Then w.p $\geq 1 - \delta$, running algorithm 1 with parameters $\gamma, \eta, T, n, \sigma, k_1, \ldots, k_d$ on distribution $\mathcal{G}_d$ returns hypothesis $h$ s.t $P_{(\boldsymbol{X}, y) \sim \mathcal{G}_d}(h(\boldsymbol{X}) \neq y) = 0$, and runs in time $O\left(d(C^2 k^2 + T) + k\|\mathbf{W}^*\|^2\right)$.*

## 6 EXPERIMENTS

A common criticism of theoretical results on deep learning is that they fail to account for the empirical success of deep networks. Indeed, negative results show that learning deep networks is computationally hard, while in practice efficient algorithms like SGD achieve remarkably good performance. Positive results, on the other hand, often make very strong assumptions that clearly do not hold in practice, like assuming the inputs are sampled from Gaussian distribution, or that they are linearly separable. Other positive results make less restrictive assumptions, but analyze algorithms that are very far from algorithms that are used in practice. At first glance, our work may seem to suffer from the same common drawbacks of positive theoretical results: we describe a distribution that is admittedly synthetic, and for deep models we analyze an algorithm that seems "tailored" to

| Classifier | Accuracy(FC) | Accuracy(Linear) |
|---|---|---|
| CNN | **0.759** | 0.735 |
| CNN(Random) | 0.645 | 0.616 |
| Clustering+JL | 0.586 | 0.588 |
| Ours | **0.734** | 0.689 |

Figure 4: Results of various configurations on the CIFAR-10 dataset

learn this distribution. To show that our model and algorithm do capture properties of distributions and algorithms used in practice, we implemented our algorithm to learn a CNN on the CIFAR-10 dataset, comparing it to a vanilla CNN trained with a common SGD-based optimization algorithm.

As our aim is to show that our algorithm achieves comparable result to a vanilla SGD-based optimization, and not to achieve state-of-the-art results on CIFAR-10, we do not use any of the common "tricks" that are widely used when training deep networks (such as data augmentation, dropout, batch normalization, scheduled learning rate, averaging of weights across iterations etc.). We implemented our algorithm by repeating the following steps twice: (1) Sample $N$ patches of size 3x3 uniformly from the dataset. (2) For some $\ell$, run the K-means algorithm to find $\ell$ cluster centers $c_1 \ldots c_\ell$. (3) At this step, we need to associate each cluster with a vector in $\mathbb{R}^\ell$, such that the image of this mapping is a set of orthonormal vectors, and then map every patch in every image to the vector corresponding to the cluster it belongs to. We do so by performing Conv3x3 operation with the $\ell$ kernels $c_1 \ldots c_\ell$, and then perform ReLU operation with a fixed bias $b$. This roughly maps each patch to the vector $e_i$, where $i$ is the cluster the patch belongs to. (4) While our analysis corresponds to performing the convolution from the previous step with a stride of 3, to make the architecture closer to the commonly used CNNs (specifically the one suggested in the Tensorflow implementation Google-Brain (2016)), we used a stride of 1 followed by a 2x2 max-pooling. (5) Randomly initialize a two layered linear network, where the first layer is Conv1x1 with $\ell'$ output channels, and the second layer is a fully-connected Affine layer that outputs 10 channels to predict the 10 classes of CIFAR-10. (6) Train the two-layers with Adam optimization (Kingma & Ba (2014)) on the cross-entropy loss, and remove the top layer. The output of this block is the output of these steps.

Repeating the above steps twice yields a network with two blocks of Conv3x3-ReLU-Pool-Conv1x1. We feed the output of these steps to a final classifier that is trained with Adam on the cross entropy loss for 100k iterations, to output the final classification of this model. We test two choices for this classifier: a linear classifier and a three-layers fully-connected neural network. Note that in both cases, the output of our algorithm is a vanilla CNN. The only difference is in the training algorithm. To calibrate the various parameters that define the model, we first perform random parameter search, where we use 10k examples from the train set as validation set (and the rest 40k as train set). After we found the optimal parameters for all the setups we compare, we then train the model again with the calibrated parameters on all the train data, and plot the accuracy on the test data every 10k iterations. The parameters found in the parameter search are listed in appendix E.

We compared our algorithm to several alternatives. First, the standard CNN configuration in the Tensorflow implementation with two variants: CNN+(FC+ReLU)[3] is the Tensorflow architecture and CNN+Linear is the Tensorflow architecture where the last three fully connected layers were replaced by a single fully connected layer. The goal of this comparison is to show that the performance of our algorithm is in the same ballpark as that of vanilla CNNs. Second, we use the same two architectures mentioned before, but while using random weights for the CNN and training only the FC layers. Some previous analyses of the success of CNN claimed that the power of the algorithm comes from the random initialization (see Daniely (2017)), and only the training of the last layer matters. As is clearly seen, random weights are far from the performance of vanilla CNNs. Our last experiment aims at showing the power of the two layer training in our algorithm (step 6). To do so, we compare our algorithm to a variant of it, in which step 6 is replaced by random projections (based on Johnson-Lindenstrauss lemma). We denote this variant by Clustering+JL. As can be seen, this variant gives drastically inferior results, showing that the training step of Conv1x1 is crucial, and finds a "good" embedding for the process that follows, as is suggested by our theoretical analysis. A summary of all the results is given in figure 4.

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

## A   PROOF OF LEMMA 1 AND THEOREM 1

Denote $m = m_0 s$ the size of the images sampled from $\mathcal{G}_1$ (these images are in $\mathbb{R}^{m \times m}$). As in section 4, we will consider the "reshaped" version of images sampled from $\mathcal{G}_1$, so we will have $\boldsymbol{X} \in \mathbb{R}^{s^2 \times m_0^2}$. Equivalently, patches of the sampled images, that belong to sets $\{S_c^{(0)}\}_{c \in \mathcal{C}}$, will be vectors in $\mathbb{R}^{s^2}$. Finally, we denote $(\boldsymbol{Z}, y) \sim \mathcal{G}_0$ the latent images in the generative process, where $\boldsymbol{Z} \in \mathcal{C}_0^{m_0 \times m_0}$. Here we will instead consider these images as vectors in $\mathcal{C}_0^{m_0^2}$. Similarly, the labeled frequency matrix of class $c$ is $\boldsymbol{F}_c \in \mathbb{R}^{m_0 \times m_0}$, and we will consider it to be a vector $\boldsymbol{F}_c \in \mathbb{R}^{m_0^2}$. In the proof we use the notations $\boldsymbol{F}_c$ to denote the negative labeled frequency matrix:

$$\boldsymbol{F}_c := \mathbb{E}_{(\boldsymbol{Z},y) \sim \mathcal{G}_i} [-y \mathcal{I}_c(\boldsymbol{Z})]$$

We will start by proving Lemma 1:

**Proof** of Lemma 1: For simplicity, we can assume $\mathcal{C}_0 = [C]$, for some natural $C$. Therefore, the linear separator of $\mathcal{G}_0$ is a matrix $\boldsymbol{W}^* \in \mathbb{R}^{C \times m_0^2}$. For $(\boldsymbol{Z}, y) \sim \mathcal{G}_0$ define $\langle \boldsymbol{W}^*, \boldsymbol{Z} \rangle = \sum_{c \in [C]} \langle \boldsymbol{W}_c^*, \mathcal{I}_c(\boldsymbol{Z}) \rangle$, where $\mathcal{I}_c$ is the operator that replaces each entry of $\boldsymbol{Z}$ with the boolean 1 if it equals $c$, or 0 otherwise. Notice that the definition of linear separability in section 3 is equivalent to $y \langle \boldsymbol{W}^*, \boldsymbol{Z} \rangle \geq 1$ for every $(\boldsymbol{Z}, y) \sim \mathcal{G}_0$.

Denote $\overline{\boldsymbol{P}}_1, \ldots, \overline{\boldsymbol{P}}_C \in \mathbb{R}^{s^2}$ such that $\overline{\boldsymbol{P}}_i = \sum_{\boldsymbol{P} \in S_i^{(0)}} \boldsymbol{P}$. From the orthonormality of the patches, for every $\boldsymbol{P} \in S_i^{(0)}$ we have $\boldsymbol{P} \cdot \overline{\boldsymbol{P}}_j = \boldsymbol{1}_{i=j}$. Now, denote $T : \mathbb{R}^{s^2 \times m_0^2} \to \mathbb{R}$ such that:

$$T(\boldsymbol{X}) = \sum_{j \in [C]} \langle \boldsymbol{W}_j^*, \overline{\boldsymbol{P}}_j^\top \boldsymbol{X} \rangle$$

For $(\boldsymbol{X}, y) \sim \mathcal{G}_1$, denote $\boldsymbol{Z}$ the latent representation of $\boldsymbol{X}$, then by the definition of the generative distribution it is easy to verify that $\mathcal{I}_j(\boldsymbol{Z}) = \overline{\boldsymbol{P}}_j^\top \boldsymbol{X}$, and therefore:

$$T(\boldsymbol{X}) = \sum_{j \in [C]} \langle \boldsymbol{W}_j^*, \mathcal{I}_j(\boldsymbol{Z}) \rangle$$

Since $T$ is a linear function such that $\|\boldsymbol{T}\|_F^2 = k \|\boldsymbol{W}^*\|_F^2$, and using standard results on the SVM algorithm, the conclusion of the lemma follows.

∎

In the rest of this section, we will prove Theorem 2. We use the notations $\boldsymbol{w}_i^{(t)}, \boldsymbol{k}_i^{(t)}$ to denote the $i$-th columns of $\boldsymbol{W}_t, \boldsymbol{K}_t$ respectively.

For some class $c \in \mathcal{C}_0$ and for some patch $\boldsymbol{x}' \in S_c^{(0)}$, denote $f_{\boldsymbol{x}'}$ a function that takes a matrix $\boldsymbol{X}$ and returns a vector $f_{\boldsymbol{x}'}(\boldsymbol{X})$ such that the $i$'th element of $f_{\boldsymbol{x}'}(\boldsymbol{X})$ is the 1 if the $i$'th column of $\boldsymbol{X}$, denoted $\boldsymbol{x}_i$, equals to $\boldsymbol{x}'$ and 0 otherwise. That is,

$$f_{\boldsymbol{x}'}(\boldsymbol{X}) := \begin{bmatrix} \boldsymbol{1}_{\boldsymbol{x}_1 = \boldsymbol{x}'} \\ \vdots \\ \boldsymbol{1}_{\boldsymbol{x}_m = \boldsymbol{x}'} \end{bmatrix}$$

Notice that from the orthonormality of the observed columns of $\boldsymbol{X}$ it follows that: $f_{\boldsymbol{x}'}(\boldsymbol{X}) = \boldsymbol{X}^\top \boldsymbol{x}'$.

We begin with proving the following technical lemma.

**Lemma 2** *For each $c \in \mathcal{C}_0$ and for each $\boldsymbol{x}' \in S_c^{(0)}$ we have:*

$$\mathbb{E}_{(\boldsymbol{X},y) \sim \mathcal{G}_1} [-y f_{\boldsymbol{x}'}(\boldsymbol{X})] = \frac{1}{k} \boldsymbol{F}_c$$

**Proof** Denote $\mathcal{D}_{\boldsymbol{Z}}$ the distribution of $(\boldsymbol{X}^{(1)}, y) \sim \mathcal{G}_1$ conditioned on $\boldsymbol{X}^{(0)} = \boldsymbol{Z}$ (the distribution of the images generated from the latent image $\boldsymbol{Z}$). Observe that

$$\mathbb{E}_{(\boldsymbol{X},y) \sim \mathcal{G}_1} [-y f_{x'}(\boldsymbol{X})] = \frac{1}{2} \sum_{y = \pm 1} -y \mathbb{E}_{\boldsymbol{Z} \sim \mathcal{D}_y} [\mathbb{E}_{X \sim \mathcal{D}_{\boldsymbol{Z}}} [f_{\boldsymbol{x}'}(\boldsymbol{X})]]$$

Recall that for each $c \in \mathcal{C}_0$, we have $|S_c^{(0)}| = k$. Therefore, for each $j \in [m_0^2]$ we have:

$$
\begin{aligned}
\frac{1}{2} \sum_{y=\pm 1} -y \mathbb{E}_{z \sim \mathcal{D}_y} \left[ \mathbb{E}_{X \sim \mathcal{D}_Z} \left[ f_{x'}(X)_j \right] \right] &= \frac{1}{2} \sum_{y=\pm 1} -y \mathbb{E}_{Z \sim \mathcal{D}_y} \left[ \mathbb{E}_{X \sim \mathcal{D}_Z} \left[ \mathbb{1}_{x_j = x'} \right] \right] \\
&= \frac{1}{2} \sum_{y=\pm 1} -y \mathbb{E}_{Z \sim \mathcal{D}_y} \left[ P_{X \sim \mathcal{D}_Z} (x_j = x') \right] \\
&= \frac{1}{2} \sum_{y=\pm 1} -y \mathbb{E}_{Z \sim \mathcal{D}_y} \left[ \frac{1}{k} \mathbb{1}_{z_j = c} \right] \\
&= \frac{1}{k} \cdot \frac{1}{2} \sum_{y=\pm 1} -y \mathbb{E}_{Z \sim \mathcal{D}_y} \left[ \mathcal{I}_c(Z)_j \right] = \frac{1}{k} [F_c]_j
\end{aligned}
$$

∎

The next lemma reveals a surprising connection between the gradient and the vectors $F_c$.

**Lemma 3** *for every $c \in \mathcal{C}_0$ and for every $x' \in S_c^{(0)}$:*

$$
x' \frac{\partial}{\partial k_i} \mathcal{L}_{K,W} = \frac{1}{k} w_i \cdot F_c
$$

**Proof** For a fixed $X$ and $W$, denote $\hat{y}(K) = \mathcal{N}_{K,W}(X)$. Note that:

$$
\frac{\partial}{\partial k_i} \hat{y}(K) = X w_i
$$

So for $x' \in S_c^{(0)}$ we have:

$$
x' \cdot \frac{\partial}{\partial k_i} \hat{y} = (x')^\top X w_i = w_i \cdot f_{x'}(X)
$$

Combining the above with the definition of the loss function, $\ell_y(\hat{y}) = -y\hat{y}$, and with Lemma 2 we get:

$$
\begin{aligned}
x' \frac{\partial}{\partial k_i} \mathcal{L}_{K_t, W_0} &= \mathbb{E}_{(X,y) \sim \mathcal{G}_1} \left[ x' \frac{\partial}{\partial k_i} \ell_y(\hat{y}) \right] \\
&= \mathbb{E}_{(X,y) \sim \mathcal{G}_1} \left[ -y \, x' \frac{\partial}{\partial k_i} \hat{y} \right] \\
&= \mathbb{E}_{(X,y) \sim \mathcal{G}_1} \left[ -y \, w_i^{(0)} \cdot f_{x'}(X) \right] \\
&= w_i^{(0)} \cdot \mathbb{E}_{(X,y) \sim \mathcal{G}_1} \left[ -y f_{x'}(X) \right] \\
&= \frac{1}{k} w_i^{(0)} \cdot F_c
\end{aligned}
$$

∎

As an immediate corollary we obtain that a gradient step does not change the projection of the kernel on two vectors that correspond to the same class (both are in the same $S_c^{(0)}$).

**Corollary 1** *For every $t \geq 0$, $i \in [n]$, for every semantic class $c \in \mathcal{C}$ and for every $x_1, x_2 \in S_c^{(0)}$ it holds that: $|k_i^{(t+1)} \cdot x_1 - k_i^{(t+1)} \cdot x_2| = |k_i^{(t)} \cdot x_1 - k_i^{(t)} \cdot x_2|$.*

**Proof** From Lemma 3 we can conclude that for a given $c \in \mathcal{C}$, for every $x_1, x_2 \in S_c^{(0)}$ we get:

$$
x_1 \frac{\partial}{\partial k_i} \mathcal{L}_{K_t, W_0} = x_2 \frac{\partial}{\partial k_i} \mathcal{L}_{K_t, W_0}
$$

From the gradient descent update rule:

$$\boldsymbol{k}_i^{(t+1)} = \boldsymbol{k}_i^{(t)} - \eta \frac{\partial}{\partial \boldsymbol{k}_i} \mathcal{L}_{\boldsymbol{K}_t, \boldsymbol{W}_0}$$

And therefore:

$$|\boldsymbol{k}_i^{(t+1)} \cdot \boldsymbol{x}_1 - \boldsymbol{k}_i^{(t+1)} \cdot \boldsymbol{x}_2| = |(\boldsymbol{k}_t^{(i)} - \eta \frac{\partial}{\partial \boldsymbol{k}_i} \mathcal{L}_{\boldsymbol{K}_t, \boldsymbol{W}_0}) \cdot \boldsymbol{x}_1 - (\boldsymbol{k}_i^{(t)} - \eta \frac{\partial}{\partial \boldsymbol{k}_i} \mathcal{L}_{\boldsymbol{K}_t, \boldsymbol{W}_0}) \cdot \boldsymbol{x}_2|$$

$$= |\boldsymbol{k}_i^{(t)} \cdot \boldsymbol{x}_1 - \boldsymbol{k}_i^{(t)} \cdot \boldsymbol{x}_2 - (\eta \frac{\partial}{\partial \boldsymbol{k}_i} \mathcal{L}_{\boldsymbol{K}_t, \boldsymbol{W}_0} \boldsymbol{x}_1 - \eta \frac{\partial}{\partial \boldsymbol{k}_i} \mathcal{L}_{\boldsymbol{K}_t, \boldsymbol{W}_0} \boldsymbol{x}_2)|$$

$$= |\boldsymbol{k}_i^{(t)} \cdot \boldsymbol{x}_1 - \boldsymbol{k}_i^{(t)} \cdot \boldsymbol{x}_2|$$

∎

Next we turn to show that a gradient step improves the separation of vectors coming from different semantic classes.

**Lemma 4** *Fix $c_1, c_2 \in \mathcal{C}_0$. Recall that we denote $\angle(\boldsymbol{F}_{c_1}, \boldsymbol{F}_{c_2})$ to be the angle between the vectors $\boldsymbol{F}_{c_1}, \boldsymbol{F}_{c_2}$. Then, with probability $\angle(\boldsymbol{F}_{c_1}, \boldsymbol{F}_{c_2})/\pi$ on the initialization of $\boldsymbol{w}_i^{(0)}$ we get:*

$$\text{sign}(\boldsymbol{w}_i^{(0)} \cdot \boldsymbol{F}_{c_1}) \neq \text{sign}(\boldsymbol{w}_i^{(0)} \cdot \boldsymbol{F}_{c_2})$$

**Proof** Observe the projection of $\boldsymbol{w}_i^{(0)}$ on the plane spanned by $\boldsymbol{F}_{c_1}, \boldsymbol{F}_{c_2}$. Then, the result is immediate from the symmetry of the initialization of $\boldsymbol{w}_i^{(0)}$. ∎

**Lemma 5** *Fix $c_1 \neq c_2 \in \mathcal{C}_0$. Then, with probability of at least $0.23 \frac{\angle(\boldsymbol{F}_{c_1}, \boldsymbol{F}_{c_2})}{\pi}$ we get for every $\boldsymbol{x}_1 \in S_{c_1}^{(0)}, \boldsymbol{x}_2 \in S_{c_2}^{(0)}$:*

$$|\boldsymbol{k}_i^{(T)} \cdot \boldsymbol{x}_1 - \boldsymbol{k}_i^{(T)} \cdot \boldsymbol{x}_2| > \frac{1}{k} \eta T \frac{\|\boldsymbol{F}_{c_1}\| + \|\boldsymbol{F}_{c_2}\|}{2} - 2\sigma$$

**Proof** Notice that since $\boldsymbol{w}_i^{(0)} \sim \mathcal{N}(0,1)$, we get that $\boldsymbol{w}_i^{(0)} \cdot \boldsymbol{F}_{c_j} \sim \mathcal{N}(0, \|\boldsymbol{F}_{c_j}\|^2)$ for $j \in \{1, 2\}$. Therefore, the probability that $\boldsymbol{w}_i^{(0)} \cdot \boldsymbol{F}_{c_j}$ deviates by at most $\frac{1}{2}$-std from the mean is $\text{erf}(\frac{1}{2\sqrt{2}})$. Thus, we get that:

$$P(|\boldsymbol{w}_i^{(0)} \cdot \boldsymbol{F}_{c_j}| \leq \|\boldsymbol{F}_{c_j}\|) = \text{erf}(\frac{1}{2\sqrt{2}})$$

And using the union bound:

$$P(|\boldsymbol{w}_i^{(0)} \cdot \boldsymbol{F}_{c_1}| \leq \|\boldsymbol{F}_{c_1}\| \vee |\boldsymbol{w}_i^{(0)} \cdot \boldsymbol{F}_{c_2}| \leq \|\boldsymbol{F}_{c_2}\|) \leq 2\text{erf}(\frac{1}{2\sqrt{2}}) < 0.77$$

Thus, using Lemma 4, we get that the following holds with probability of at least $0.23 \frac{\angle(\boldsymbol{F}_{c_1}, \boldsymbol{F}_{c_2})}{\pi}$:

- $|\boldsymbol{w}_i^{(0)} \cdot \boldsymbol{F}_{c_1}| > \|\boldsymbol{F}_{c_1}\|$
- $|\boldsymbol{w}_i^{(0)} \cdot \boldsymbol{F}_{c_2}| > \|\boldsymbol{F}_{c_2}\|$
- $\text{sign}(\boldsymbol{w}_i^{(0)} \cdot \boldsymbol{F}_{c_1}) \neq \text{sign}(\boldsymbol{w}_i^{(0)} \cdot \boldsymbol{F}_{c_2})$

Assume w.l.o.g that $\boldsymbol{w}_i^{(0)} \cdot \boldsymbol{F}_{c_1} < 0 < \boldsymbol{w}_i^{(0)} \cdot \boldsymbol{F}_{c_2}$, then using Lemma 3 we get:

$$\boldsymbol{k}_i^{(T)} \boldsymbol{x}_1 = \boldsymbol{k}_i^{(0)} \boldsymbol{x}_1 - \eta \sum_{t=1}^{T} \frac{\partial}{\partial \boldsymbol{k}^{(i)}} \boldsymbol{x}_1 \mathcal{L}_{\boldsymbol{K}_t, \boldsymbol{W}_0}$$

$$= \boldsymbol{k}_i^{(0)} - \eta \sum_{t=1}^{T} \frac{1}{k} \boldsymbol{w}_i^{(0)} \cdot \boldsymbol{F}_{c_1}$$

$$= \boldsymbol{k}_i^{(0)} - \frac{1}{k} \eta T \boldsymbol{w}_i^{(0)} \cdot \boldsymbol{F}_{c_1} > \frac{1}{k} \eta T \frac{\|\boldsymbol{F}_{c_1}\|}{2} - \sigma$$

In a similar fashion we can get:

$$\boldsymbol{k}_i^{(T)} \boldsymbol{x}_2 < -\frac{1}{k}\eta T \frac{\|\boldsymbol{F}_{c_2}\|}{2} + \sigma$$

And thus the conclusion follows:

$$\boldsymbol{k}_i^{(T)} \boldsymbol{x}_1 - \boldsymbol{k}_i^{(T)} \boldsymbol{x}_2 > \frac{1}{k}\eta T \frac{\|\boldsymbol{F}_{c_1}\| + \|\boldsymbol{F}_{c_2}\|}{2} - 2\sigma$$

■

Finally, we are ready to prove the main theorem.

**Proof** of Theorem 1.
We show two things:

1. Fix $c \in \mathcal{C}_0$. By the initialization, we get that for every $\boldsymbol{x}_1, \boldsymbol{x}_2 \in S_c^{(0)}$ and for every $i \in [n]$:

$$|\boldsymbol{x}_1 \cdot \boldsymbol{k}_i^{(0)} - \boldsymbol{x}_2 \cdot \boldsymbol{k}_i^{(0)}| < \frac{\sigma}{\sqrt{n}}$$

Using Corollary 1, we get that:

$$|\boldsymbol{x}_1 \cdot \boldsymbol{k}_i^{(T)} - \boldsymbol{x}_2 \cdot \boldsymbol{k}_i^{(T)}| < \frac{\sigma}{\sqrt{n}}$$

And thus:

$$\|\boldsymbol{K}_T \boldsymbol{x}_1 - \boldsymbol{K}_T \boldsymbol{x}_2\| < \sigma$$

2. Let $c_1 \neq c_2 \in \mathcal{C}_0$. Assume $T > \frac{(\gamma + 2\sigma)k}{\eta\lambda}$. For $i \in [n]$, from Lemma 5 we get that with probability of at least $0.23\frac{\angle(\boldsymbol{F}_{c_1}, \boldsymbol{F}_{c_2})}{\pi} > 0.23\frac{\theta}{\pi}$ for every $\boldsymbol{x}_1 \in S_{c_1}^{(0)}, \boldsymbol{x}_2 \in S_{c_2}^{(0)}$:

$$|\boldsymbol{k}_i^{(T)} \boldsymbol{x}_1 - \boldsymbol{k}_i^{(T)} \boldsymbol{x}_2| > \frac{1}{k}\eta T \frac{\|\boldsymbol{F}_{c_1}\| + \|\boldsymbol{F}_{c_2}\|}{2} - 2\sigma > \frac{1}{k}\eta T\lambda - 2\sigma > \gamma$$

For a given $c_1 \neq c_2 \in \mathcal{C}_0$, denote the event:

$$A_{c_1,c_2} = \{\forall i \in [n] : |\boldsymbol{k}_i^{(T)} \cdot \boldsymbol{x}_1 - \boldsymbol{k}_i^{(T)} \cdot \boldsymbol{x}_2| < \gamma, \ \boldsymbol{x}_1 \in S_{c_1}^{(0)}, \boldsymbol{x}_2 \in S_{c_2}^{(0)}\}$$

Then, from what we have showed, it holds that:

$$P(A_{c_1,c_2}) < (1 - 0.23\frac{\theta}{\pi})^n \leq \exp(-0.23n\frac{\theta}{\pi})$$

Using the union bound, we get that:

$$P(\exists c_1 \neq c_2 \in \mathcal{C}_0 \ s.t \ A_{c_1,c_2}) < \exp(-0.23n\frac{\theta}{\pi})|\mathcal{C}_0|^2$$

Choosing $n > \frac{2\pi}{0.23\theta}\log(\frac{|\mathcal{C}|}{\delta})$ we get $P(\exists c_1 \neq c_2 \in \mathcal{C}_0 \ s.t \ A_{c_1,c_2}) < \delta$. Now, if for every $c_1 \neq c_2 \in \mathcal{C}_0$ the event $A_{c_1,c_2}$ doesn't hold, then clearly for every $\boldsymbol{x}_1 \in S_{c_1}^{(0)}, \boldsymbol{x}_2 \in S_{c_2}^{(0)}$ we would get $\|\boldsymbol{K}_T \cdot \boldsymbol{x}_1 - \boldsymbol{K}_T \cdot \boldsymbol{x}_2\| > \gamma$, and this is what we wanted to show.

■

## B  ANALYSIS OF SGD

We show a theorem equivalent to Theorem 1, when using the SGD algorithm on samples from distribution $\mathcal{G}_1$, instead of updating on the population gradient with the GD algorithm. For this analysis, we fix $m_0 = 3$ and take the columns of $\boldsymbol{W}$ to be initialized on the unit sphere.

**Theorem 3** *Assume that assumptions 2, 3, 4 hold. Assume we train a two-layer network of size $n > \frac{2\pi}{0.12\theta} \log(\frac{|\mathcal{C}|}{\delta})$ with SGD with batch size of 1 on samples from distribution $\mathcal{G}_1$, for $T > \max\{\frac{(\gamma+4\sigma)^4 k^4}{(0.4\lambda)^4}, \frac{8n^2 m_0^4}{\sigma^4} \log^2(\frac{2|\mathcal{C}_0|kn}{\delta})\}$ iterations, with learning rate $\eta = T^{-3/4}$, for some $\gamma > 0$. Then with probability of at least $1 - 2\delta$:*

1. *for each $c \in \mathcal{C}_0$, for every $x_1, x_2 \in S_c^{(0)}$ we get $\|\mathbf{K}_T^\top \cdot x_1 - \mathbf{K}_T^\top \cdot x_2\| < 2\sigma$*

2. *for $c_1, c_2 \in \mathcal{C}_0$, if $c_1 \neq c_2$, for every $x_1 \in S_{c_1}^{(0)}, x_2 \in S_{c_2}^{(0)}$, we get $\|\mathbf{K}_T^\top \cdot x_1 - \mathbf{K}_T^\top \cdot x_2\| > \gamma$*

Assume we are given a sample $(\mathbf{X}_1, y_1), \ldots, (\mathbf{X}_T, y_T)$ sampled i.i.d from distribution $\mathcal{G}_1$, and we run SGD with batch size of 1 (with respect to the same problem analyzed in the previous section). Denote by $\hat{y}_t$ the prediction of the network on iteration $t$. Denote $\mathbf{g}_i^{(t)}$ the gradient with respect to the kernel $\mathbf{k}_i$ on iteration $t$.

Similar to Lemma 3, we get for every $c \in \mathcal{C}_0$ and $\mathbf{x}' \in S_c^{(0)}$:

$$\mathbf{x}' \mathbf{g}_i^{(t)} = \mathbf{x}' \frac{\partial}{\partial \mathbf{k}_i} \ell_{y_t}(\hat{y}_t) = -y_t \mathbf{x}' \frac{\partial}{\partial \mathbf{k}_i} \mathbf{w}_i^{(0)} \cdot f_{x'}(\mathbf{X})$$

Using Lemma 3 along with the fact that $(\mathbf{X}_t, y_t) \sim_{i.i.d} \mathcal{G}_1$ we get that:

$$\mathbb{E}_{(\mathbf{X}_t, y_t) \sim \mathcal{G}_1} \left[ \mathbf{x}' \mathbf{g}_i^{(t)} \right] = \frac{1}{k} \mathbf{w}_i^{(0)} \cdot \mathbf{F}_c$$

Since we assume $\|\mathbf{w}_0^{(i)}\| \leq 1$, we get that:

$$|\eta \mathbf{x}' \mathbf{g}_i^{(t)}| \leq \eta \|\mathbf{w}_i^{(0)}\| \|f_{x'}(\mathbf{X}_t)\| \leq \eta m_0$$

Therefore, fixing $\alpha > 0$, and using Hoeffding's bound we get:

$$Pr(|\sum_{t=1}^{T} \eta \mathbf{x}' \mathbf{g}_i^{(t)} - \frac{\eta T}{k} \mathbf{w}_i^{(0)} \cdot \mathbf{F}_c| \geq \alpha) \leq 2 \exp(-2\alpha^2/(T\eta^2 m_0^2))$$

And taking $\eta = T^{-3/4}$ we get:

$$Pr(|\sum_{t=1}^{T} \eta \mathbf{x}' \mathbf{g}_i^{(t)} - \frac{\eta T}{k} \mathbf{w}_i^{(0)} \cdot \mathbf{F}_c| \geq \alpha) \leq 2 \exp(-2\sqrt{T}\alpha^2/m_0^2)$$

Denote by $A$ the event where:

$$\exists i \in [n], \exists \mathbf{x}' \in \cup_{c \in \mathcal{C}_0} S_c^{(0)} : |\sum_{t=1}^{T} \eta \mathbf{x}' \mathbf{g}_i^{(t)} - \frac{T}{k} \mathbf{w}_i^{(0)} \cdot \mathbf{F}_c| \geq \alpha$$

Taking $T \geq \frac{m_0^4}{2\alpha^4} \log^2(\frac{2|\mathcal{C}_0|kn}{\delta})$ and using union bound over all kernels $k_i, i \in [n]$ and patches in $\cup_{c \in \mathcal{C}_0} S_c^{(0)}$ we get:

$$Pr(A) \leq \delta$$

Using the above, we get the following:

**Corollary 2** *With probability at least $1 - \delta$, we get:*

$$|\mathbf{k}_i^{(T)} \cdot \mathbf{x}_1 - \mathbf{k}_i^{(T)} \cdot \mathbf{x}_2| \leq \frac{\sigma}{\sqrt{n}} + 2\alpha$$

**Proof** With probability at least $1 - \delta$, for every $c \in \mathcal{C}_0$ and for every $x_1, x_2 \in S_c^{(0)}$ we get:

$$|\boldsymbol{k}_i^{(T)} \cdot \boldsymbol{x}_1 - \boldsymbol{k}_i^{(T)} \cdot \boldsymbol{x}_2| = |\boldsymbol{k}_i^{(0)} \cdot \boldsymbol{x}_1 - \boldsymbol{k}_i^{(0)} \cdot \boldsymbol{x}_2 - \eta \sum_{t=1}^{T} \boldsymbol{x}' \cdot \boldsymbol{g}_i^{(t)} + \eta \sum_{t=1}^{T} \boldsymbol{x}' \cdot \boldsymbol{g}_i^{(t)}|$$

$$\leq |\boldsymbol{k}_i^{(0)} \cdot \boldsymbol{x}_1 - \boldsymbol{k}_i^{(0)} \cdot \boldsymbol{x}_2| + |\sum_{t=1}^{T} \eta \boldsymbol{x}_1 \boldsymbol{g}_i^{(t)} - \frac{T}{k} \boldsymbol{w}_i^{(0)} \cdot \boldsymbol{F}_c|$$

$$+ |\sum_{t=1}^{T} \eta \boldsymbol{x}_2 \boldsymbol{g}_i^{(t)} - \frac{T}{k} \boldsymbol{w}_i^{(0)} \cdot \boldsymbol{F}_c|$$

$$\leq \frac{\sigma}{\sqrt{n}} + 2\alpha$$

We repeat the proof of Lemma 5, now assuming the updates are of SGD. ∎

**Lemma 6** *Assume event $A$ does not occur. Fix $c_1 \neq c_2 \in \mathcal{C}_0$. Then, with probability of at least $0.23 \frac{\angle(\boldsymbol{F}_{c_1}, \boldsymbol{F}_{c_2})}{\pi}$ we get for every $\boldsymbol{x}_1 \in S_{c_1}^{(0)}, \boldsymbol{x}_2 \in S_{c_2}^{(0)}$:*

$$|\boldsymbol{k}_i^{(T)} \cdot \boldsymbol{x}_1 - \boldsymbol{k}_i^{(T)} \cdot \boldsymbol{x}_2| > \eta T (\frac{1}{k} \frac{\|\boldsymbol{F}_{c_1}\| + \|\boldsymbol{F}_{c_2}\|}{2} - 2\alpha) - 2\sigma$$

**Proof** Denote $v := \frac{1}{2}(\langle \boldsymbol{w}_i^{(0)}, \boldsymbol{F}_{c_j}\rangle/\|\boldsymbol{F}_{c_j}\| + 1)$. The vector $\boldsymbol{w}_i^{(0)}$ is distributed uniformly on the sphere, and from spherical symmetry we can see that the distribution of $v$ is independent of the value of $\boldsymbol{F}_{c_j}/\|\boldsymbol{F}_{c_j}\|$. Therefore, we can see that the distribution of $v$ is simply the distribution of the first coordinate of $\boldsymbol{w}_i^{(0)}$, normalized to $[0, 1]$. Since $w_0^{(0)}$ is of dimension $m_0^2$, we get that $v \sim \text{Beta}((m_0^2 - 1)/2, (m_0^2 - 1)/2)) = \text{Beta}(4, 4)$. Therefore, we get:

$$P(0.4 \leq v \leq 0.6) < 0.44$$

And from this we get:

$$P(|\boldsymbol{w}_i^{(0)} \cdot \boldsymbol{F}_{c_j}| \leq 0.2\|\boldsymbol{F}_{c_j}\|) < 0.44$$

And using the union bound:

$$P(|\boldsymbol{w}_i^{(0)} \cdot \boldsymbol{F}_{c_1}| \leq 0.2\|\boldsymbol{F}_{c_1}\| \vee |\boldsymbol{w}_i^{(0)} \cdot \boldsymbol{F}_{c_2}| \leq 0.2\|\boldsymbol{F}_{c_2}\|) < 0.88$$

Thus, using Lemma 4, we get that the following holds with probability of at least $0.12 \frac{\angle(\boldsymbol{F}_{c_1}, \boldsymbol{F}_{c_2})}{\pi}$:

- $|\boldsymbol{w}_i^{(0)} \cdot \boldsymbol{F}_{c_1}| > 0.2\|\boldsymbol{F}_{c_1}\|$
- $|\boldsymbol{w}_i^{(0)} \cdot \boldsymbol{F}_{c_2}| > 0.2\|\boldsymbol{F}_{c_2}\|$
- $\text{sign}(\boldsymbol{w}_i^{(0)} \cdot \boldsymbol{F}_{c_1}) \neq \text{sign}(\boldsymbol{w}_i^{(0)} \cdot \boldsymbol{F}_{c_2})$

Assume w.l.o.g that $\boldsymbol{w}_i^{(0)} \cdot \boldsymbol{F}_{c_1} < 0 < \boldsymbol{w}_i^{(0)} \cdot \boldsymbol{F}_{c_2}$, then using what we have shown:

$$\boldsymbol{k}_i^{(T)} \boldsymbol{x}_1 = \boldsymbol{k}_i^{(0)} \boldsymbol{x}_1 - \sum_{t=1}^{T} \eta \boldsymbol{x}_1 \boldsymbol{g}_i^{(t)}$$

$$= \boldsymbol{k}_i^{(0)} \boldsymbol{x}_1 - \sum_{t=1}^{T} \eta \boldsymbol{x}_1 \boldsymbol{g}_i^{(t)} + \frac{\eta T}{k} \boldsymbol{w}_i^{(0)} \cdot \boldsymbol{F}_{c_1} - \frac{\eta T}{k} \boldsymbol{w}_i^{(0)} \cdot \boldsymbol{F}_{c_1}$$

$$\geq \boldsymbol{k}_i^{(0)} x_1 - \frac{\eta T}{k} \boldsymbol{w}_i^{(0)} \cdot \boldsymbol{F}_{c_1} - |\frac{1}{T} \sum_{t=1}^{T} \boldsymbol{x}_1 \boldsymbol{g}_i^{(t)} - \frac{\eta T}{k} \boldsymbol{w}_i^{(0)} \cdot \boldsymbol{F}_{c_1}|$$

$$> \frac{0.2T^{1/4}}{k} \|\boldsymbol{F}_{c_1}\| - \sigma - \alpha$$

In a similar fashion we can get:

$$k_i^{(T)} x_2 < -\frac{0.2T^{1/4}}{k} \|F_{c_2}\| + \sigma + \alpha$$

And thus the conclusion follows:

$$k_i^{(T)} x_1 - k_i^{(T)} x_2 > \frac{0.2T^{1/4}}{k}(\|F_{c_1}\| + \|F_{c_2}\|) - 2\sigma - 2\alpha$$

$\blacksquare$

We can now repeat the main theorem for the SGD case:

**Proof** of Theorem 3.

Assume event $A$ does not occur, and fix $\alpha = \frac{\sigma}{2\sqrt{n}}$. We show two things:

1. Using Corollary 2, we get that:

$$|x_1 \cdot k_i^{(T)} - x_2 \cdot k_i^{(T)}| < \frac{2\sigma}{\sqrt{n}}$$

   And thus:

$$\|K_T x_1 - K_T x_2\| < 2\sigma$$

2. Let $c_1 \neq c_2 \in \mathcal{C}_0$. Assume $T > \frac{(\gamma+4\sigma)^4 k^4}{(0.4\lambda)^4}$. For $i \in [n]$, from Lemma 6 we get that with probability of at least $0.12\frac{\angle(F_{c_1}, F_{c_2})}{\pi} > 0.12\frac{\theta}{\pi}$ for every $x_1 \in S_{c_1}^{(0)}, x_2 \in S_{c_2}^{(0)}$:

$$|k_i^{(T)} x_1 - k_i^{(T)} x_2| > \frac{0.2T^{1/4}}{k}(\|F_{c_1}\| + \|F_{c_2}\|) - 2\sigma - 2\alpha > \frac{0.4T^{1/4}}{k}\lambda - 4\sigma > \gamma$$

   For a given $c_1 \neq c_2 \in \mathcal{C}_0$, denote the event:

$$A_{c_1,c_2} = \{\forall i \in [n] : |k_i^{(T)} \cdot x_1 - k_i^{(T)} \cdot x_2| < \gamma, \ x_1 \in S_{c_1}^{(0)}, x_2 \in S_{c_2}^{(0)}\}$$

   Then, from what we have showed, it holds that:

$$P(A_{c_1,c_2}) < (1 - 0.12\frac{\theta}{\pi})^n \leq \exp(-0.12n\frac{\theta}{\pi})$$

   Using the union bound, we get that:

$$P(\exists c_1 \neq c_2 \in \mathcal{C}_0 \ s.t \ A_{c_1,c_2}) < \exp(-0.12n\frac{\theta}{\pi})|\mathcal{C}_0|^2$$

   Choosing $n > \frac{2\pi}{0.12\theta}\log(\frac{|\mathcal{C}|}{\delta})$ we get $P(\exists c_1 \neq c_2 \in \mathcal{C}_0 \ s.t \ A_{c_1,c_2}) < \delta$. Now, if for every $c_1 \neq c_2 \in \mathcal{C}_0$ the event $A_{c_1,c_2}$ doesn't hold, then clearly for every $x_1 \in S_{c_1}^{(0)}, x_2 \in S_{c_2}^{(0)}$ we would get $\|K_T \cdot x_1 - K_T \cdot x_2\| > \gamma$, and this is what we wanted to show.

Since event $A$ happens with probability at most $\delta$, the conclusion follows. $\blacksquare$

## C    PROOF OF THEOREM 2

Our algorithm uses the k-means++ algorithm, which is a variant of Lloyd's algorithm where the initial cluster centers are chosen with probability proportional to their distance to the closest cluster that was already chosen. The algorithm first chooses a cluster center $c_1$ uniformly on all examples. Any new cluster is chosen in the following way: assume we already chose cluster centers $c_1, \ldots, c_n$, denote $D(x) := \min_{j \in [n]} \|x - c_j\|$, the minimal distance from example $x \in S$ to the closest cluster center. Then, the algorithm chooses an example $x \in S$ with probability $\frac{D(x)^2}{\sum_{x' \in S} D(x')^2}$. After the cluster centers are chosen, we run the standard Lloyd's algorithm for k-means.

The following lemma shows that k-means++ finds an optimal solution with high probability on highly-clustered data:

**Lemma 7** *Fix $\delta > 0$. Let $S$ be a finite set that is "highly-clustered": $S$ is partitioned such that $S = \cup_{j \in [C]} B_j$, where the partition satisfies:*

1. *For every $\boldsymbol{x}, \boldsymbol{y} \in B_j$ it holds that $\|\boldsymbol{x} - \boldsymbol{y}\| < 1$.*

2. *For every $\boldsymbol{x} \in B_i, \boldsymbol{y} \in B_j$ such that $i \neq j$ it holds that $\|\boldsymbol{x} - \boldsymbol{y}\| > C/\sqrt{\delta}$.*

3. *All $B_j$-s are sets of fixed size: $\forall_{i,j} |B_i| = |B_j|$.*

*Assume we run the k-means++ algorithm on set $S$ to find $C$ clusters. Then with probability at least $1 - \delta$, the algorithm returns an "optimal clustering": the centers returned by the algorithm $\hat{\boldsymbol{x}}_1, \ldots, \hat{\boldsymbol{x}}_C$ satisfy that for every $j \in [C]$ there exist $i \in [C]$ with $\hat{\boldsymbol{x}}_i \in B_j$.*

**Proof** We will prove by induction that at the $i$-th step of choosing the centers, with probability at least $1 - \frac{i}{C}\delta$, the chosen centers $\hat{\boldsymbol{x}}_1, \ldots, \hat{\boldsymbol{x}}_i$ satisfy that there are no $i_1 \neq i_2 \in [i]$ such that $\hat{\boldsymbol{x}}_{i_1}, \hat{\boldsymbol{x}}_{i_2} \in B_j$:

- for $i = 1$ this is immediate.

- assume we chose $\hat{\boldsymbol{x}}_1, \ldots, \hat{\boldsymbol{x}}_i$ that satisfy the above condition, and w.l.o.g we can assume that for $j \leq i$ we have $\hat{\boldsymbol{x}}_j \in B_j$. Then, for every $\boldsymbol{x} \in B_j$ with $j \leq i$ we have $D(\boldsymbol{x}) < 1$, and for every $\boldsymbol{x} \in B_j$ with $j > i$ we have $D(\boldsymbol{x}) > C/\sqrt{\delta}$. Therefore, the probability of choosing $\boldsymbol{x} \in B_j$ with $j \leq i$ in the next step is at most:

$$\frac{i}{i + (C - i)C^2/\delta} \leq \frac{C - 1}{C^2/\delta} \leq \frac{\delta}{C}$$

  Now, using the union bound we get the required.

Taking $i = C$ proves the initialized center are already optimal. Clearly, any step of Lloyd's algorithm will maintain this property. In fact, the algorithm will converge after one step, returning $\hat{\boldsymbol{x}}_1, \ldots, \hat{\boldsymbol{x}}_C$ with $\hat{\boldsymbol{x}}_j = \frac{1}{C} \sum_{\boldsymbol{x} \in B_j} \boldsymbol{x}$. ∎

Before we complete the proof of the theorem, we remind a few notations that were used in the algorithm's description. We use $\phi_i$ to denote the clustering of patches learned in the $i$-th iteration of the algorithm, and $K_i$ the weights of the kernels learned BEFORE the $i$-th step (thus, the patches mapped by $K_i$ are the input to the clustering algorithm that outputs $\phi_i$). Note that we do not learn a mapping in the last step, as from Lemma 1 as the distribution $\mathcal{G}_1$ is linearly separable, so we could simply use SVM on the output of the network at that stage. Finally, we use the notations $\phi * A$ to indicate that we operate $\phi$ on every patch of the tensor $A$. When we use operations on distributions, for example $h \circ \mathcal{G}$ or $\phi * \mathcal{G}$, we refer to the new distribution generated by applying these operation to every examples sampled from $\mathcal{G}$. The essence of the proof is the following lemma:

**Lemma 8** *Let $\mathcal{G} := \mathcal{G}_d$ be the distribution over pairs $(\boldsymbol{X}, y)$, where $\boldsymbol{X}$ is the observed image over the reals, and recall that for $i < d$, the distribution $\mathcal{G}_i$ is over pairs $(\boldsymbol{X}^{(i)}, y)$ where $\boldsymbol{X}^{(i)}$ is in a space of latent semantic images over $\mathcal{C}_i$. For every $i \in [d]$, with probability at least $1 - 2(d - i)\delta'$, there exists an orthonormal patch mapping $\varphi_i : \mathcal{C}_i^{s \times s} \to \mathbb{R}^{k_i}$ such that $\phi_i * (h_i \circ \mathcal{G}) = \varphi_i * \mathcal{G}_i$, where $\phi_i$ and $h_i$ are as defined in algorithm 1.*

The lemma tells us that the neural network at step $i$ of the algorithm reveals (in some sense) the latent semantic structure.

We will again use a "reshaped" notation: in every level $i$ (the $i$-th step of the induction), we treat patches $\boldsymbol{P} \in \mathcal{C}_i^{s \times s}$ as vectors $\boldsymbol{P} \in \mathcal{C}_i^{s^2}$. For a sub-image in the next level, denoted $\boldsymbol{Z} \in \mathcal{C}_i^{s^2 \times s^2}$, we will denote $\boldsymbol{Z}_j \in \mathcal{C}_{i+1}^{s^2}$ to be the $j$-th patch of size $s \times s$ in the sub-image (generated by the class at the $j$-th coordinate of higher level patch $\boldsymbol{P} \in \mathcal{C}_i^{s^2}$).

We will prove by induction the following claim:

**Lemma 9** *Assume that for some $i < d$ the condition of the lemma holds for $i + 1$, namely that there exists an orthonormal patch mapping $\varphi_{i+1} : \mathcal{C}_{i+1}^{s \times s} \to \mathbb{R}^{k_{i+1}}$ such that $\phi_{i+1} * (h_{i+1} \circ \mathcal{G}) = \varphi_{i+1} * \mathcal{G}_{i+1}$. Then, with probability at least $1 - 2\delta'$ this condition holds for $i$.*

**Proof** Let $\varphi_{i+1}$ be the mapping satisfying the condition of the claim for $i + 1$. Notice that the data that is fed to the two-layer training step comes from the distribution $\varphi_{i+1} * \mathcal{G}_{i+1}$, and satisfies the conditions for the analysis of Theorem 1.

First, we will show that the set of patches $S_i$, the set of all patches in the distribution $h_i \circ \mathcal{G}$ is "highly-clustered" with probability at least $1 - \delta'$.

For every $c \in \mathcal{C}_i$, denote $B_c = \{K_i^\top \varphi_{i+1}(\boldsymbol{P}) : \boldsymbol{P} \in S_c^{(i)}\}$. Recall that from our assumption, $h_i \circ \mathcal{G} = K_i^\top(\varphi_{i+1} * \mathcal{G}_{i+1})$. Therefore, we have that $S_i = \cup_{c \in \mathcal{C}_i} B_c$. Now, from Theorem 1, since we have $T > \frac{k(C/\sqrt{\delta'} + 2\sigma)}{\eta\lambda}$ and $\sigma = \frac{1}{s}$, with probability at least $1 - \delta'$, conditions 1 and 2 in the "highly-clustered" definition hold. Since we assume that $S_c^{(i)}$ are of the same size, condition 3 follows.

From the above, using Lemma 7, with probability at least $1 - 2\delta'$ the set $S_i$ is "highly-clustered" and k-means++ returns an optimal clustering. We will now limit ourselves to the event that both of these hold.

Now, define the map $\varphi_i : \mathcal{C}_i^{s \times s} \to \mathbb{R}^k$ in the following way: first, for every patch $\boldsymbol{P} \in \mathcal{C}_i^{s^2}$ we take an arbitrary manifestation of the patch $\boldsymbol{P}$ in the next level, denoted $\boldsymbol{Z} \in \mathcal{C}_{i+1}^{s^2 \times s^2}$. In other words, $\boldsymbol{Z}$ could be any $s^2 \times s^2$ sub-image in the next level that could be generated from the patch $\boldsymbol{P}$. Now, take $\varphi_i(\boldsymbol{P}) := \phi_i(K_i \cdot (\varphi_{i+1} * \boldsymbol{Z}))$. Then, the following holds:

1. $\varphi_i(\boldsymbol{P})$ does not depend on the choice of $\boldsymbol{Z}$: if $\boldsymbol{Z}, \boldsymbol{Z}'$ are two different manifestations of $\boldsymbol{P}$, then, from the definition of the generative model, for every $j \in [s]$ it holds that $\boldsymbol{Z}_j, \boldsymbol{Z}'_j \in S_{\boldsymbol{P}_j}$. Thus from what we have shown:

$$\|K_i \cdot \varphi_{i+1}(\boldsymbol{Z}_j) - K_i \cdot \varphi_{i+1}(\boldsymbol{Z}'_j)\| < \sigma = \frac{1}{s}$$

   Therefore:

$$\|K_i \cdot \varphi_{i+1} * \boldsymbol{Z} - K_i \cdot \varphi_{i+1} * \boldsymbol{Z}'\|^2 = \sum_{j=1}^{s^2} \|K_i \cdot \varphi_{i+1}(\boldsymbol{Z}_j) - K_i \cdot \varphi_{i+1}(\boldsymbol{Z}'_j)\|^2$$
$$< 1$$

   and since we get an optimal clustering, we have:

$$\phi_i(K_i \cdot (\varphi_{i+1} * \boldsymbol{Z})) = \phi_i(K_i \cdot (\varphi_{i+1} * \boldsymbol{Z}'))$$

2. for every two patches $\boldsymbol{P} \neq \boldsymbol{P}'$ we get $\varphi_\kappa(\boldsymbol{P}) \perp \varphi_\kappa(\boldsymbol{P}')$: let $\boldsymbol{Z}, \boldsymbol{Z}'$ be the manifestations of $p, p'$ respectively. Since $\boldsymbol{P} \neq \boldsymbol{P}'$ there exists $j \in [s^2]$ such that $\boldsymbol{P}_j \neq \boldsymbol{P}'_j$. From the generative model it follows that $\boldsymbol{Z}_j \in S_{\boldsymbol{P}_j}^{(i)}$ and $\boldsymbol{Z}'_j \in S_{\boldsymbol{P}'_j}^{(i)}$, and therefore from the behavior of the algorithm: $\|K_i^\top \cdot \varphi_{i+1}(\boldsymbol{Z}_j) - K_i^\top \cdot \varphi_{i+1}(\boldsymbol{Z}'_j)\| > C/\sqrt{\delta'} > k_i/\sqrt{\delta'}$. Therefore, we get that $\|K_i \cdot \varphi_{i+1} * \boldsymbol{Z} - K_i \cdot \varphi_{i+1} * \boldsymbol{Z}'\| > k_i/\sqrt{\delta}$, and thus from the clustering optimality we get $\varphi_i(\boldsymbol{P}) \perp \varphi_i(\boldsymbol{P}')$.

Now, recall that in the algorithm definition: $h_i = K_i^\top \cdot (\phi_{i+1} * h_{i+1})$. Using the assumption for $i + 1$, we get:

$$h_i \circ \mathcal{G} = K_i(\varphi_{i+1} * \mathcal{G}_{i+1})$$

and from the definition of $\varphi_i$ and what we have shown we get:

$$\phi_i * (h_i \circ \mathcal{G}) = \varphi_i * \mathcal{G}_i$$

■

**Proof** [of Lemma 8] We prove by induction:

- Note that an immediate property of the k-means++ algorithm is that it doesn't choose the same example twice. Therefore, if we run k-means++ on sample $S$ to find $|S|$ clusters, we

get a single cluster center for each element of $S$. From the definition of our model, there are $|\mathcal{C}_{d-1}|k$ patches in the observed image, and since we run k-means++ to find $k_d = |\mathcal{C}_{d-1}|k$ clusters, we get a cluster center for each patch. Hence, k-means++ returns an orthogonal patch mapping $\phi_d$, and since $h_d = id$ we get the required.

- Assume the claim holds for $i + 1$, then with probability at least $1 - 2(d - i - 1)\delta'$ the condition holds for $i + 1$. In such event, from Lemma 9 with probability at least $1 - 2\delta'$, it holds for $i$. Therefore, with probability $(1 - 2(d - i - 1)\delta')(1 - 2\delta') \geq 1 - 2(d - i)\delta'$ it holds for $i$.

■

**Proof** [ of Theorem 2] From Lemma 8, after performing $d$ iterations of the algorithm, we observe that with probability at least $1 - 2d\delta' = 1 - \delta$ distribution $\varphi_1 * \mathcal{G}_1$, where $\varphi_1$ is an orthonormal patch mapping. Therefore, using Lemma 1, it is linearly separable with margin $k\|\mathbf{W}^*\|^2$. Therefore, the SVM algorithm finds an optimal separator in $O(k\|\mathbf{W}^*\|^2)$. To count the overall amount of iterations, notice that the when running k-means++ on sample $S$ to find $C$ clusters, the initialization step at most $|S|C$ iterations and the algorithm itself converges after one iteration, so in our case its runtime is bounded by $C^2 k^2$. The runtime of the $TLGD$ algorithm is $T$. We perform $d$ iterations of clustering and the $TLGD$ algorithm, and finally run the SVM algorithm that converges in $O(k\|\mathbf{W}^*\|^2)$. So the runtime of the whole algorithm is $O\left(d(C^2 k^2 + T) + k\|\mathbf{W}^*\|^2\right)$.

■

# D    SYNTHETIC DATA EXPERIMENT

To confirm that our model is typically not linearly separable, we generated synthetic examples using our model in the following way:

- We chose $\mathcal{Y} = \{\pm 1\}$.
- We fixed $\mathcal{C}_0 = \{0, 1, 2, 3\}$, and chose 4 vectors for each class, uniformly over $\mathcal{C}_0^{3 \times 3}$. Notice that overall there are 8 vectors labeled with $\pm 1$, and they are linearly independent with high probability (we also validated that they are indeed independent). This process defines the distribution $\mathcal{G}_0$, and it is therefore indeed linearly separable.
- We fix $\mathcal{C}_1 = \{0, 1, 2, 3\}$, and for each $c \in \mathcal{C}_0$ we choose the set $S_c^{(0)}$ be drawing $k$ vectors uniformly over $\mathcal{C}_1^{3 \times 3}$. This defines the distribution $\mathcal{G}_1$.
- Finally, we choose $\mathcal{C}_2 = \mathbb{R}^3$, and for each $c \in \mathcal{C}_1$ we choose $S_c^{(1)}$ by drawing $k$ vectors from a normal distribution over $\mathcal{C}_2^{3 \times 3}$. This defines the distribution $\mathcal{G}_2$, which is the observed distribution.

We run this model with different values of $k$ $(30, 40, 50)$. We then measure the values of $\theta$ and $\lambda$, to show that Assumption 3 holds for this distribution.

We generate a sample of $50,000$ examples from this distribution, taking $40,000$ as train examples, and $10,000$ as test. We train a linear classifier on this distribution, and display its performance on the test data. For comparison, we also train a CNN with non-overlapping convolutions (convolutions of 3x3 and stride 3) with the ReLU activation. We use an architecture of two layers of convolution, and a final readout layer, and train with respect to the logistic loss. This training is done with the Adam optimizer in a standard end-to-end fashion. We also train the same architecture with our algorithm, where we the training is done layer-by-layer.

# E    PARAMETERS FOR CIFAR-10 EXPERIMENTS

figure 5 below lists the parameters that were learned in the random parameter search for the different configurations of the algorithm, as described in 4. The table lists the parameters used in each layer: $\ell_1, \ell_2$ are the number of clusters for the first and second layer, and $\ell_1', \ell_2'$ are the output channels of the Conv1x1 operation for each layer. These parameters could be used to reproduce the results of our experiments.

| Classifier | $N$ | $k_1$ | $k_1'$ | $k_2$ | $k_2'$ | $b$ | Accuracy |
|---|---|---|---|---|---|---|---|
| Ours+FC | 47509 | 1377 | 155 | 3534 | 216 | -0.14 | **0.734** |
| Ours+Linear | 32124 | 1384 | 97 | 2576 | 211 | -0.63 | 0.689 |
| Clustering+JL+FC | 12369 | 39 | 39 | 184 | 184 | 0.84 | 0.586 |
| Clustering+JL+Linear | 57893 | 5004 | 345 | 6813 | 407 | -0.01 | 0.588 |

Figure 5: Parameters used in our experiment
.

