# OpenReview forum: "Provable Guarantees on Learning Hierarchical Generative Models with Deep CNNs"
_ICLR.cc/2019/Conference_

### Official Review · AnonReviewer2 · 2018-11-01
**Unsupported claims; assumptions should be more elaborated**

**Rating:** 4
**Confidence:** 4

**Review:**

The paper claims to propose a computationally efficient algorithm for training deep CNNs by making assumptions about the distribution of data. The authors argue that (i) they don't make very simplistic assumptions about the data generating distribution as some other papers do, and (ii) their algorithm resembles the actual methods that are used for training deep models and shows some surprising properties of SGD.

Throughout the paper, the authors make a number of assumptions which seem arbitrary at times; not much justifications are provided. The authors claim that their assumptions are not as simplistic as assuming e.g., the inputs are sampled from Gaussian distribution. Unfortunately this is highly unclear: while the "assumptions" themselves are complex, the combination of those assumptions may make the problem solution trivial. While proving a lower bound to address this issue may be hard, at least the authors should try to illuminate more why the solution is not trivial (e.g., why a linear classifier doesn't work, etc.)

Despite the claims, I find the proposed algorithm very far from the usual SGD-based training methods; this is not a problem per se but I don't think that the result illuminates on the effectiveness of SGD (as the authors suggest). The proposed algorithm is a greedy layer-wise method that in each level does a clustering and also trains a "linear" CNN with SGD. So the hardness of end-to-end training of a deep network does not show up. Furthermore, it is not clear for training a linear CNN the SGD is even needed.

I suggest that the authors name each of the assumptions and clearly say which ones are assumed for which result. Here are some of the assumptions that the authors talk about.

1_ The data is generated by the following recursive procedure: First a small "high-level image" is generated from a distribution, G_0. The "pixels" of this high-level image are supposed to encode semantic classes, e.g., sky or ground. In the next step, each of these high-level pixels are turned into a small (lower-level) image. Therefore, we will have a more refined image after the second step. (each semantic class (e.g., sky) has a corresponding distribution that generates the smaller lower-level image (e.g., uniform over 4 possible types of skies)). This procedure continues recursively until we have the final image.

2_ G_0 is "linearly separable".

3_ Semantic classes defined in the model are different enough from each other

4_ {F_c} corresponding to semantic classes are linearly independent

5_ Patch Orthonormality (apparently not assumed everywhere)


it appears that if one assumes all of 1-5, then the problem becomes trivial (linearly separable). The authors then say that we don't want to make assumption 5 for this reason; still, the problem solution may be trivial (authors should at least intuitively justify why it isn't )

Here are some more uses of the word "assumption".

6_ "For simplicity of analysis, we assume only the first layer of the network is trained".

7_ "We assume the algorithm [KMEANS++] returns a mapping [...] such that [...]"

The experiments do not seem conclusive. Only a few experiments have been done. I think the acquired results for CIFAR-10 are below the usual ones using CNNs, and the effects of various hyper-parameters may have interfered.

--
After reading the authors' response, I still think the way that the contributions are depicted (e.g., a justifying the effectiveness of SGD) are inaccurate/unsupported.

Furthermore, although the authors' suggest that they have tested a linear classifier and observed that the data is not linearly separable, more explanations/intuitions are needed about the assumptions that are made throughout the paper.

---

> ### Author Response · Authors · 2018-12-03
> **Response to Reviewer2**
>
> Thank you for the response.
> We will give here a few notes of clarification about the assumptions, and we can add these to the final revision of the paper. We hope that these comments provide the intuitions and explanations that are missing.
>
> Assumption 1: The linear separability of the latent distribution captures the fact that the observed images are generated from a latent distribution that is "simple" to learn.
>
> Assumption 2: We assume that sets of patches that belong to different semantic classes are disjoint - this is just another way of writing that the there exists a partition of the set of patches, where each subset in the partition is identified with a semantic class. The assumption that these subsets have the same size is just for simplification of the notations in the analysis.
>
> Assumption 3: We require that the frequency matrices of patches from different semantic classes are linearly independent in pairs - i.e, that two frequency matrices of different semantic classes are not linearly dependent (similar up to scaling by a positive scalar). This is a way to make sure that the semantic classes defined in the model are different from each other, as otherwise one could define many different models that generate the same output distribution. The empirical experiments that were added in Figure 3 show that this requirement holds for a model with random parameters.
>
> Assumption 4: The assumption about the orthonormality of the patches is given only in section 4, as a technical step that is not needed in the later analysis of the full algorithm.

---

### Official Review · AnonReviewer3 · 2018-11-02
**A generative model for images**

**Rating:** 6
**Confidence:** 3

**Review:**

After the rebuttal: I appreciate the authors' effort to revise the paper. The revision made clear that the data produced by the proposed generative model is not linearly separable in general while the theory (Theorem 2) still holds.

I am keeping my original evaluation as there still seems to be a lack of stronger experimental evidence. The fact that the classification algorithm motivated by the generative model can do as well as a similar-sized ConvNet does not quite support that the generative model itself is good -- getting a good classifier is still an easier task than getting a good generative model.

=====================

This paper proposes a new generative model for natural images. Based on the architecture of the generative model, a “layer-wise clustering” algorithm for image classification is proposed and theoretically shown to converge to an optimal classifier. Experimentally, the algorithm is shown to have similar performances as a baseline CNN on CIFAR-10.

The main novelty of this paper is the proposed hierarchical generative model and the associated algorithm. One interesting feature is that the network obtained by this algorithm is entirely linear except for the ReLU-pool part. However, the ReLU-pool does not serve as a typical nonlinearity / pooling I believe; rather it sounds to me like a specially tailored step for the theoretical results, which under the “patch orthonormality” assumption is guaranteed to recover the previous layer. Therefore, it surprises me a little bit that the algorithm actually works reasonably well on CIFAR-10. However, as the baseline it compares with is still below "typical", I do want to see if this algorithm can be scaled up to match the performance of more complicated (at least pre-ResNet) models such as VGG.

The theoretical result looks appealing, but I feel like the magic more or less comes from the strong assumptions. In particular, in expectation the output image is just a *linear* operator on the initial (m_0 x m_0 x C_0) one-hot semantic variable. Also, the patch orthonormality assumption implies that intermediate semantics can be perfectly recovered by the (clustering + conv with centroids + ReLU-pool) step, as we are just recovering a partition of a group of orthonormal vectors.

---

> ### Author Response · Authors · 2018-12-03
> **Response to Reviewer3**
>
> Thank you for the response, and we are happy that our revisions provided some clarifications and missing details.
> As for your concerns regarding our generative model: we find this generative model appealing not because it is interesting by itself, but because we can give provable guarantees about learning it with neural networks.
> We prove that this model can be learned with ConvNets trained by our algorithm, while giving empirical evidence that this "tailored" algorithm works on real data. The fact that a vanilla ConvNet can learn this generative model as well does not weaken our theoretical results, but only shows that the standard training algorithms for ConvNets have similar properties to our suggested algorithm. This is another evidence that our theoretical analysis captures properties that are relevant for the practice of deep learning.

---

### Official Review · AnonReviewer1 · 2018-11-03
**Interesting theoretical results under slightly unrealistic assumptions**

**Rating:** 6
**Confidence:** 3

**Review:**

The paper first puts forward a generative model for labelled images. The generative model is hierarchical and interesting although a bit complicated. They then show that when there is only one latent layer (i.e., two overall layers) in the generative model, the latent layer can be learned by gradient descent under a linear convolutional model. Inspired by this, the authors present an algorithm for the general case which involves using the two-layer algorithm iteratively to learn each individual layer of the full model. There is a theoretical result proving that this algorithm works. I find the theoretical results interesting.

It must be said though that the generative model is quite complicated and somewhat unrealistic. The theoretical results are proved under additional stringent assumptions. For example, Theorem 1 applies to Gradient Descent applied to the population loss as opposed to the actual SGD. Also, the GD in practice here is with respect to both K and W. But the analysis is restricted to the setting where W is fixed. Is it possible to prove a version of Theorem 1 that applies to the actual SGD? Further when Theorem 1 is invoked in the proof of Theorem 2 (specifically in the proof of Lemma 8), the fact that Theorem 1 applies to population loss is glossed over? I also fear that the assumptions in Theorem 2 may be too strong. The fact that one can find orthonormal patches in each layer together with the assumption that the images in the final layer are linearly separable might mean that there is some sort of linear separability overall? It would be good if the authors can clarify this. The conclusion of Theorem 2 (that the algorithm learns a hypothesis with zero error) seems too strong to me (perfect classification is usually possible only under clear separability assumptions).

The paper is also slightly hard to read with too many assumptions of various kinds floating around.

---

### Author Response · Authors · 2018-11-11
**Revision to the paper and response to all reviewers**

We thank the reviewers for their thorough review and helpful feedback. We have tried to address the main concerns raised by the reviewers:

1. Linear separability of the described distribution: a main concern of all the reviewers was that although we described a complex process of generating images, when taking into account all the different assumptions it might become a simple linearly separable distribution. We show that this is not the case, by running a linear classification algorithm on such distribution and measure its performance. We generated a distribution using our model, and indeed showed that a linear classifier achieves very poor performance (around 0.6 accuracy). See Section 3.3.

2. Both Reviewer 1 and Reviewer 3 found the assumptions that were introduced along the paper hard to follow. Following the suggestion of Reviewer 3, we enumerated our distributional assumptions, and clearly stated which ones where assumed for each theorem/lemma.

3. Analysis of the SGD algorithm: for simplicity of the analysis, we gave our theoretical results assuming the training algorithm is gradient descent with respect to the population loss. Since in practice we use SGD, Reviewer 1 asked whether we can prove a similar result for the SGD case. To address this, we have added a theorem similar to Theorem 1 for the stochastic case (SGD), and it is now given in the appendix.

---

### Meta-Review · Area_Chair1 · 2018-12-13

**Confidence:** 3
**Recommendation:** Reject

**Metareview:**

This manuscript proposes a generative model for images, then proposes a training procedure for fitting a convolutional neural network based on this model. One novelty if this result is that the generative procedure seems to be more complex than generative assumptions required for previous work. It is clear that the problem addressed -- training methods that may improve on SGD, with convergence guarantees -- is of significant interest to the community.

The reviewers and AC note several issue (i) the initial version of the manuscript includes several assumptions that are not clearly stated. This seems to have been fixed in the updated manuscript (ii) reviewers suspect that the accumulation of stated assumptions may result in an easily separable generative model -- limiting the generality of the results (iii) experiemental results are underwhelming, and only comparable to much older published results.